# Coherent theta activity in the medial and orbital frontal cortices encodes reward value

Linda M Amarante, Mark Laubach*

Department of Neuroscience, American University, Washington DC, United States

**ABSTRACT** This study examined how the medial frontal (MFC) and orbital frontal (OFC) cortices process reward information. We simultaneously recorded local field potentials in the two areas as rats consumed liquid sucrose rewards. Both areas exhibited a 4–8 Hz 'theta' rhythm that was phase-locked to the lick cycle. The rhythm tracked shifts in sucrose concentrations and fluid volumes, demonstrating that it is sensitive to differences in reward magnitude. The coupling between the rhythm and licking was stronger in MFC than OFC and varied with response vigor and absolute reward value in the MFC. Spectral analysis revealed zero-lag coherence between the cortical areas, and found evidence for a directionality of the rhythm, with MFC leading OFC. Our findings suggest that consummatory behavior generates simultaneous theta range activity in the MFC and OFC that encodes the value of consumed fluids, with the MFC having a top-down role in the control of consumption.

*For correspondence:
mark.laubach@american.edu

Competing interest: The authors declare that no competing interests exist.

## Introduction

The medial and orbital frontal cortices (MFC and OFC) are two of the most studied parts of the cerebral cortex for their role in value-guided decision making, a process that ultimately results in animals consuming rewarding foods or fluids. There are extensive anatomical connections between the various parts of the MFC and OFC in rodents (*Gabbott et al., 2003*; *Gabbott et al., 2005*; *Barreiros et al., 2020*), and the regions are part of the medial frontal network (*Öngür and Price, 2000*). The MFC and OFC are thought to have specific roles in the control of behavior and specific homologies with medial and orbital regions of the primate frontal cortex (MFC: *Laubach et al., 2018*; OFC: *Izquierdo, 2017*). The extensive interconnections between MFC and OFC suggest that the two regions work together to control value-guided decisions. Unfortunately, few, if any, studies have examined concurrent neural processing in these regions of the rodent brain as animals perform behavioral tasks that depend on the two cortical regions.

In standard laboratory tasks, the action selection and outcome evaluation phases of value-guided decisions are commonly conceived as separate processes (*Rangel et al., 2008*). MFC and OFC may contribute independently to these processes or interact concurrently across them. Though there is some variation across published studies, most argue for MFC having a role in action-outcome processing (*Alexander and Brown, 2011*; *Simon et al., 2015*) and OFC having a role in stimulus-outcome (stimulus-reward) processing (*Gallagher et al., 1999*; *Schoenbaum and Roesch, 2005*; *Simon et al., 2015*). The present study directly compared neural activity in the MFC and OFC of rats as they performed a simple consummatory task, called the shifting values licking task, or SVLT (*Parent et al., 2015a*). Importantly, the task depends on the ability of animals to guide their consummatory behavior based on the value of available rewards, and performance of these kinds of tasks depends on both the MFC (*Parent et al., 2015a*; *Parent et al., 2015b*) and OFC (*Kesner and Gilbert, 2007*). The goal of the study was to use the SVLT to determine if the MFC and OFC have distinct roles in

processing reward information, for example, varying with action (licking) in MFC and the sensory properties of the rewards in OFC.

Most published studies on reward processing used operant designs with distinct actions preceding different outcomes. For example, a rat might respond in one of two choice ports to produce a highly valued reward, delivered from a separate reward port. To collect the reward, the rat has to travel across an operant chamber and then collect a food pellet or initiate licking on a spout to collect the reward. In such tasks (*Pratt and Mizumori, 2001*; *van Duuren et al., 2009*; *van Wingerden et al., 2010*; *Riceberg and Shapiro, 2017*; *Jarovi et al., 2018*; *Siniscalchi et al., 2019*), neural activity during the period of consumption might reflect the properties of the reward, how the animal consumes it, and/or the behaviors that precede reward collection (e.g. locomotion). As such, it is difficult to isolate reward specific activity using such operant designs.

Several published studies have used simpler consummatory and Pavlovian designs, and found neural activity in the MFC is selectively modulated during active consumption (*Petykó et al., 2009*; *Horst and Laubach, 2013*; *Petykó et al., 2015*). None of these tasks used fluids with different reward values. *Amarante et al., 2017*, was the first study to examine if similar neural activity was associated with animals consuming different magnitudes of reward. The study used the SVLT and presented rats with rewards that differed in terms of the concentration of sucrose contained in the rewarding fluids. The study found that neural activity in the MFC is entrained to the animals' lick cycle and the strength of entrainment varies with the value of the rewarding fluid, that is, stronger entrainment with higher value reward. The study also used reversible inactivation methods to demonstrate that licking entrainment depends on the MFC.

In the present study, we used the SVLT, and several variations on the basic task design, to study consumption-related activity in MFC and OFC. Spectral analyses were used to account for the extent

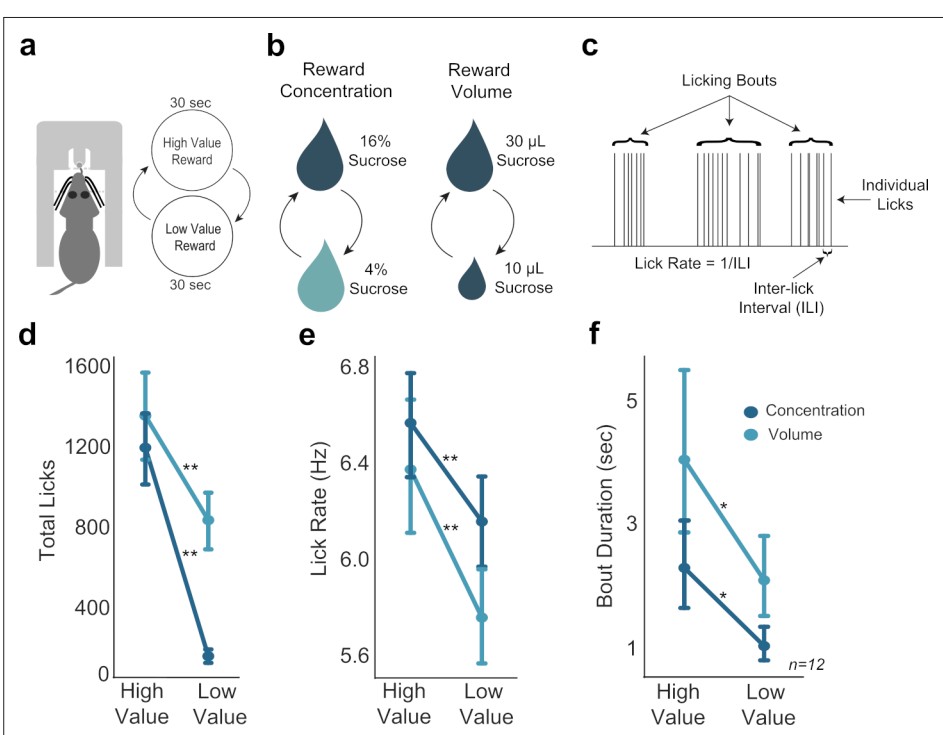

**Figure 1.** Consummatory behavior tracked shifts in sucrose concentration and fluid volume. (**a**) In the shifting values licking task, rats received access to one of two values of reward, with rewards alternating every 30 s. (**b**) Manipulation of reward value by changing either concentration or volume. (**c**) Types of behavioral licking measurements recorded in all licking tasks. (**d**,**e**,**f**) Rats licked more (**d**), faster (**e**), and over longer bouts (**f**) for the high concentration and large volume rewards. Single asterisk (*) denotes p < 0.05; double asterisk (**) denotes p < 0.001. Error bars represent 95 % confidence intervals.

The online version of this article includes the following figure supplement(s) for figure 1:

**Figure supplement 1.** Electrode localization.

to which neural activity in each area was entrained to licking and if there was evidence for direction-ality of lick entrainment among sites in the MFC and OFC. A custom-designed syringe pump was used to deliver different volumes of fluid over a common time period (*Amarante et al., 2019*). This device allowed us to directly compare neural activity associated with differences in sucrose concentration and fluid volume. We further manipulated the predictability of changes in reward magnitude to assess how predictable and unpredictable rewards are processed and used a third, intermediate level of reward to assess if reward magnitudes are encoded in a relative or absolute manner. Our findings suggest that both areas encode the value of consumed fluids and that the MFC may have a top-down role in coordinating reward processing.

## Results

### SVLT: effects of reward magnitude on consummatory behavior

The SVLT (*Amarante et al., 2017*; *Figure 1a*) was used to assess reward encoding across the MFC and OFC as 12 rats experienced shifts in reward value defined by differences in sucrose concentration or fluid volume. Shifts in concentration were between 16% and 4% sucrose in a volume of 30 µL. Shifts in volume were between 30 and 10 µL containing 16 % sucrose. Concentrations and volumes alternated over periods of 30 s (*Figure 1b*, left). Local field potential (LFP) activity was recorded from 16-channel multi-electrode arrays in the MFC in 10 of the 12 rats and OFC in 6 of the 12 rats (recording locations are shown in *Figure 1—figure supplement 1*).

Several measures of licking behavior varied with sucrose concentration or fluid volume: lick counts, inter-lick intervals, lick rate, and bout duration (*Figure 1c*). All rats licked more for the high concentration reward compared to the low concentration reward (paired t-test; t(11)=10.76, p < 0.001) (*Figure 1d*). Rats also licked at a faster rate for the high concentration reward compared to the low concentration reward (paired t-test; t(11)=6.347, p < 0.001) (*Figure 1e*). Additionally, rats had increased bout durations when licking for the high concentration reward compared to the low concentration reward (paired t-test: t(11)=2.943, p = 0.013) (*Figure 1f*). There was no difference in variability of high or low concentration licks: the coefficient of variation for inter-lick intervals was the same (paired t-test: t(9)=0.864, p = 0.41).

Rats behaved similarly when consuming the high concentration and large volume rewards. In volume manipulation sessions, rats emitted more licks for the large reward than the small reward (paired t-test; t(11)=4.99, p < 0.001). However, this difference in lick counts was less robust than the difference in high and low concentration rewards during concentration manipulation sessions (*Figure 1d*). Rats licked at a faster rate for large rewards compared to small volume rewards (paired t-test; t(11)=6.311, p < 0.001) (*Figure 1e*), and licking bouts were longer for large rewards compared to bouts to consume small rewards (*Figure 1f*), (paired t-test; t(11)=2.569, p = 0.027).

### SVLT: coherent fluctuations in the theta range in the MFC and OFC

LFPs in the MFC (N = 56) and OFC (N = 64) from four rats with arrays implanted in both cortical areas were recorded during the standard SVLT. The LFPs were analyzed with cross-correlation and a spectral method called directed coherence to assess the extent of coordinated fluctuations between the cortical regions (*Figure 2a*). Data from all rats tested with both shifts in sucrose concentration and fluid volume were used for this analysis. One of the rats had 16 LFPs recorded in each area (256 pairs). Two rats had 14 LFPs in MFC and 16 in OFC (224 pairs). The fourth rat had 12 LFPs in MFC and 16 in OFC (192 pairs). Data from a total of 896 electrode pairs were analyzed. As shown in *Figure 2b*, LFPs from both areas showed frank fluctuations during periods of sustained licking (bouts). Standard (non-directional) coherence for the LFPs peaked around a value of 0.6 near the licking frequency (*Figure 2c*). By measuring cross-correlation over a range of lags (time domain directionality), we found evidence for near zero-lag correlations. (This analysis is done in the time domain and does not account for frequency-specific directional influences.)

Directed coherence values at the licking frequency were larger for MFC leading OFC compared to OFC leading MFC (*Figure 2e*). Notably, the magnitude of the coherence was no more than 0.2, suggesting a potential weak influence of MFC on the timing of fluctuations in OFC. The magnitude of the coherences was variable over electrodes, and plots of the measures onto the anatomical

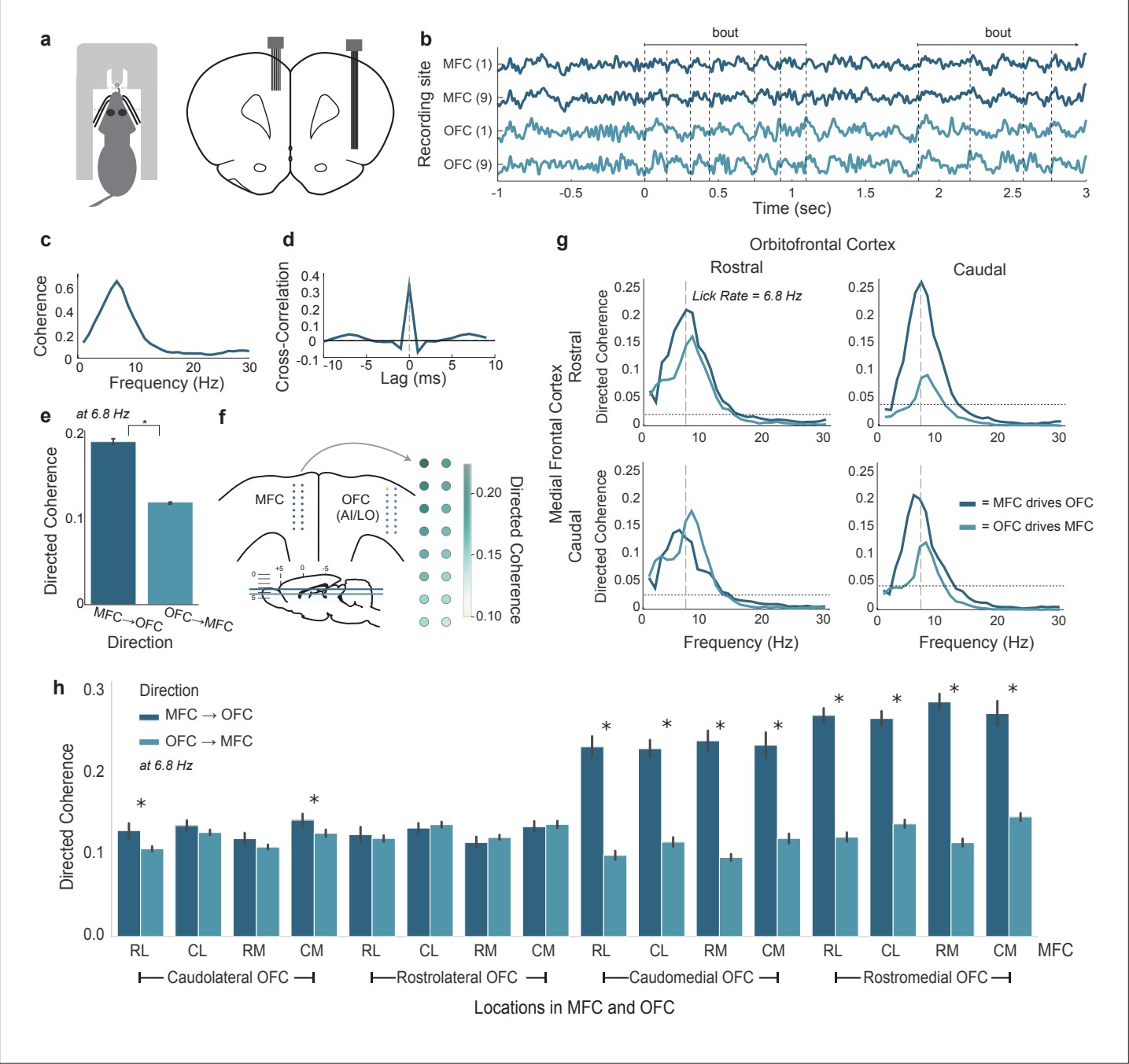

**Figure 2.** Coherent licking-related theta-band activity in the medial frontal (MFC) and orbital frontal (OFC) cortices. (**a**) Depiction of rat performing the shifting values licking task (left) and placement of recording arrays in the MFC and lateral frontal cortex (right). (**b**) Traces of simultaneous local field potential (LFP) recordings from rostral (1) and caudal (9) recording sites on arrays implanted in MFC and OFC. Two licking bouts are noted and the times of licks are shown as dashed vertical lines. (**c**) Standard (non-directional) coherence between a pair of LFPs from MFC and OFC showed a peak near the licking frequency (~7 Hz). (**d**) Cross-correlation (time domain) showed a central peak with lag near 0 ms. (e) Directed coherence at the licking frequency for MFC→OFC and OFC→MFC over all pairs of LFPs recorded in four rats. Asterisk denotes $p < 10^{-6}$ for effect of direction on coherence. (**f**) Anatomical map of directed coherence values over one of the arrays. (**g**) Directed coherence over frequencies up to 30 Hz, plotted for rostral and caudal sites in the MFC and OFC (panel **b**). (**h**) Group summary of directed coherence over all pairs of recordings. Asterisks (*) denotes $p < 0.05$. Error bars represent 95 % confidence intervals.

arrangement of the recording arrays revealed a gradient of directed coherence, with most rostral sites in MFC having larger coherence values compared to caudal sites (an example is shown in *Figure 2f*).

To further examine the role of spatial location on directed coherence, we denoted the locations of the recordings along the arrays as rostral or caudal (i.e. for each linear array with eight electrodes, the four most rostral electrodes were denoted as rostral and the rest as caudal). An example of directed coherence over frequencies up to 30 Hz for the rostral and caudal sites (*Figure 2b*) is shown in *Figure 2g*. Directed coherence was larger for the direction MFC → OFC for most rostral electrode in the MFC and both the rostral and caudal electrodes in the OFC. The caudal electrode in the MFC had larger directed coherence for the direction MFC → OFC for the caudal, but not the rostral, electrode in OFC. A group summary of these findings, at the licking frequency, is shown in *Figure 2h*. Here, the locations of the electrodes was further split as medial and lateral, and differences in directed coherence were apparent for rostral and caudal sites in the MFC and medial sites in the OFC (right half of the plot). Directed coherence was equivocal for rostral and caudal sites in the MFC and lateral sites in the OFC (left half of the plot). Based on anatomical mapping of the arrays, the medial and lateral

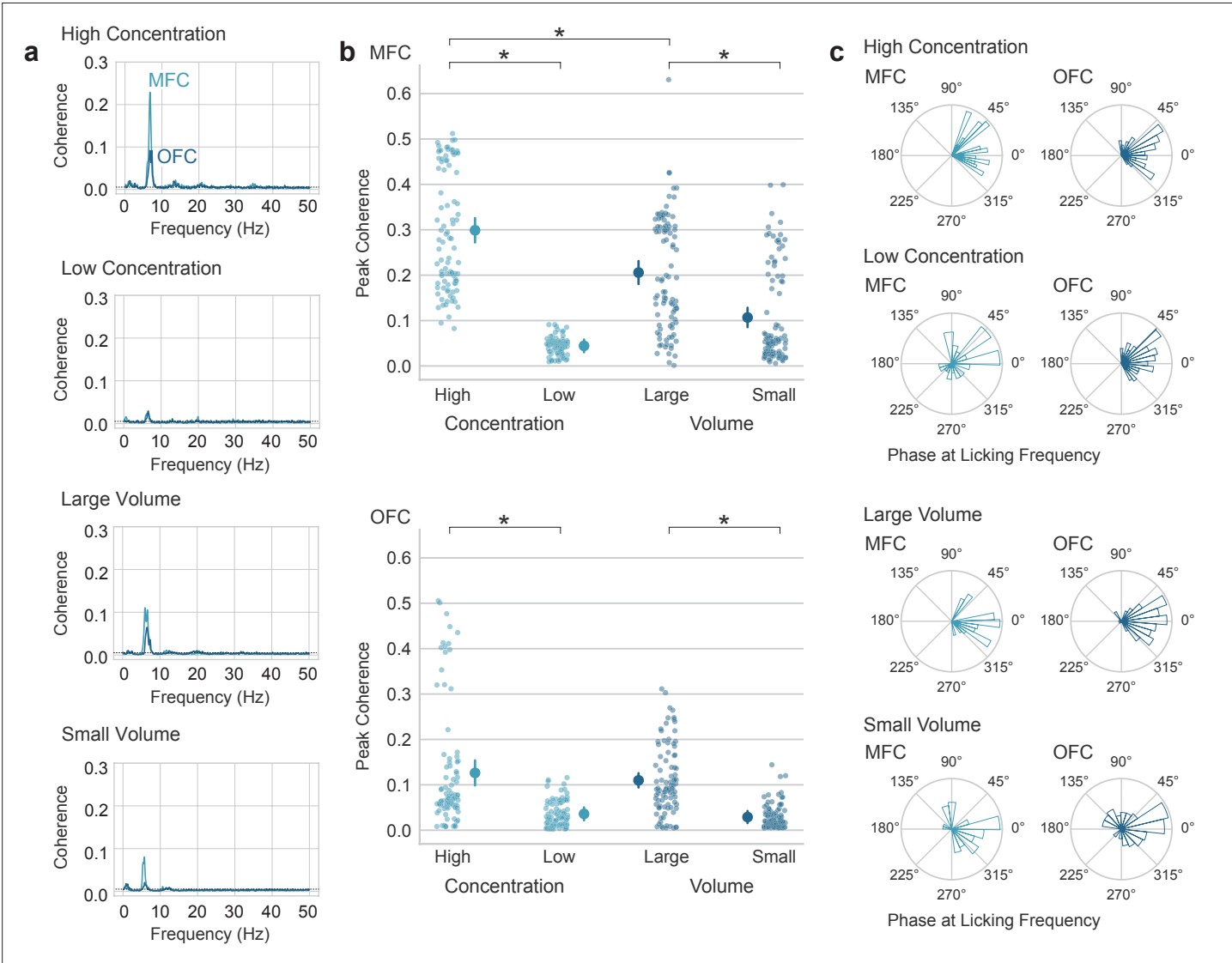

**Figure 3.** Lick entrainment in the medial frontal (MFC) and orbital frontal (OFC) cortices is sensitive to reward value. (**a**) Average lick-field coherence for local field potentials (LFPs) in MFC and OFC and licks for high and low concentration sucrose solutions and large and small fluid volumes. (**b**) Peak coherence in the range of theta (4–12 Hz) for LFPs from MFC and OFC and licks for high and low concentration sucrose solutions and large and small fluid volumes. Asterisks (*) denotes p < 0.05. Error bars represent 95 % confidence intervals. (**c**) Phase angles of LFPs at the licking frequency. Most LFPs were coherent with licks at phases between 45 and 315 degrees.

sites in the OFC were associated with the deep and superficial layers of the cortex, respectively. These findings suggest cross-laminar differences in the timing of the LFP fluctuations, with the rostral part of the MFC 'driving' fluctuations in the deep layers of the OFC, and possibly serving as feedback from the MFC to the OFC (*Gabbott et al., 2003*).

## SVLT: lick entrainment in MFC and OFC tracks reward magnitude

We next aimed to determine if there were electrophysiological differences in MFC and OFC during access to the different types of rewards. Lick-field coherence (using methods originally developed for spike-field coherence in the Neurospec library for MATLAB, *Halliday et al., 1995*). LFPs from both areas were coherent with licks at the licking frequency, and not at higher harmonic frequencies of licking (*Figure 3a*). Coherence levels were higher for licks that delivered high-value fluid (concentration and volume) compared to low-value fluid in MFC (paired t-test; concentration: t(95)=39.972, p < 0.001; volume: t(95)=11.643, p < 0.001) and OFC (paired t-test: concentration: t(91)=17.386, p < 0.001; volume: t(91)=18.970, p < 0.001) (*Figure 3a, b*). Furthermore, coherence was higher for high-value licks in the concentration shift sessions compared to the volume shift sessions in MFC (paired t-test; t(95)=6.901, p < 0.001), but not the OFC (paired t-test; t(91)=-0.401, p = 0.688). Phase angles at the licking frequency are shown in *Figure 3c*. With lick-field coherence ranging between 0 and 0.5, this analysis suggests that the LFP fluctuations at the licking frequency are only partially accounted for by the animals' licking behavior and the extent of entrainment differs between cortical areas (larger in MFC) and is sensitive to reward value (larger for higher value fluid).

Three additional measurements of LFP activity were examined: amplitude (as measured by the size of event-related potentials (ERP); *Figure 4—figure supplement 1a*), spectral power (as measured by event-related spectral power (ERSP); *Figure 4—figure supplement 1b*), and phase (as measured by inter-trial coherence (ITC), *Figure 4—figure supplement 1c*). Similar to results from lick-field coherence, we found lick-entrained activity in MFC and OFC that varied with both differences in sucrose concentration and fluid volume (*Figure 4*). ERPs showed evidence for time-locked rhythmic fluctuations in LFPs from both cortical areas (*Figure 4b and f*). Both cortical areas showed elevated ITC between 4 and 8 Hz for licks that delivered the high concentration liquid sucrose but not the low concentration sucrose (*Figure 4c and g*). That is, the phase angles of the LFP fluctuations at the times of licks were more consistent when rats consumed the high concentration fluid compared to the low concentration fluid. This result was observed in all rats that were tested (dark blue lines in *Figure 4d and h*) (MFC: F(1,278)=443, p < 0.001; OFC: F(1,177)=77.31, p < 0.001; one-way ANOVAs with an error term for within-subject variation). Analysis of phase coherence (*Figure 4—figure supplement 1d*) and event-related power (*Figure 4—figure supplement 1e*) revealed effects solely in the 4–8 Hz (theta) frequency range.

To assess differences in power, we used a peak-to-peak analysis of ERPs during licks for the high-value and low-value rewards. The measure calculates the difference in the maximum and minimum ERP amplitude using a window centered around each lick. The size of the window was twice each rat's median inter-lick interval. LFPs in MFC showed increased amplitudes for high concentration rewards, as opposed to low concentration rewards (one-way ANOVA: F(1,278)=34.19, p < 0.001). *Figure 4b* shows MFC ERPs for high and low concentration rewards of an example rat. This effect was not significant in OFC ERPs, as seen in *Figure 4f* (F(1,177)=0.557, p = 0.456). We also measured ERSP, and although there was a decrease in MFC power from licks for the high to low concentration rewards specifically in the 4–8 Hz range (F(1,278)=18.72, p < 0.001; one-way ANOVA), post hoc testing revealed no relevant significance between high and low concentration licks (p = 0.413). There was no major difference in ERSP measures in OFC (F(1,177)=0.039, p = 0.843).

In sessions with shifts in fluid volume, ERPs in MFC or OFC did not distinguish between large versus small volume rewards (MFC: F(1,216)=0.865, p = 0.354; OFC: (F(1,179)=1.876, p = 0.173); one-way ANOVAs (*Figure 4b and f*, bottom). There was no major difference in ERSP during licks for large or small rewards in MFC or OFC (MFC: F(1,216)=0.877, p = 0.35; OFC: F(1,179)=1.76, p = 0.186); one-way ANOVAs). However, in both MFC and OFC, rats showed similar 4–8 Hz phase locking for large rewards (*Figure 4c and g*, bottom), closely resembling what we observed with high concentration rewards (*Figure 4c and g* top). Phase locking was significantly increased for small rewards (MFC: F(1,216)=138.5, p < 0.001; OFC: F(1,179)=280.8, p < 0.001; one-way ANOVA) and was observed in all rats that were tested (light blue lines in *Figure 4d and h*).

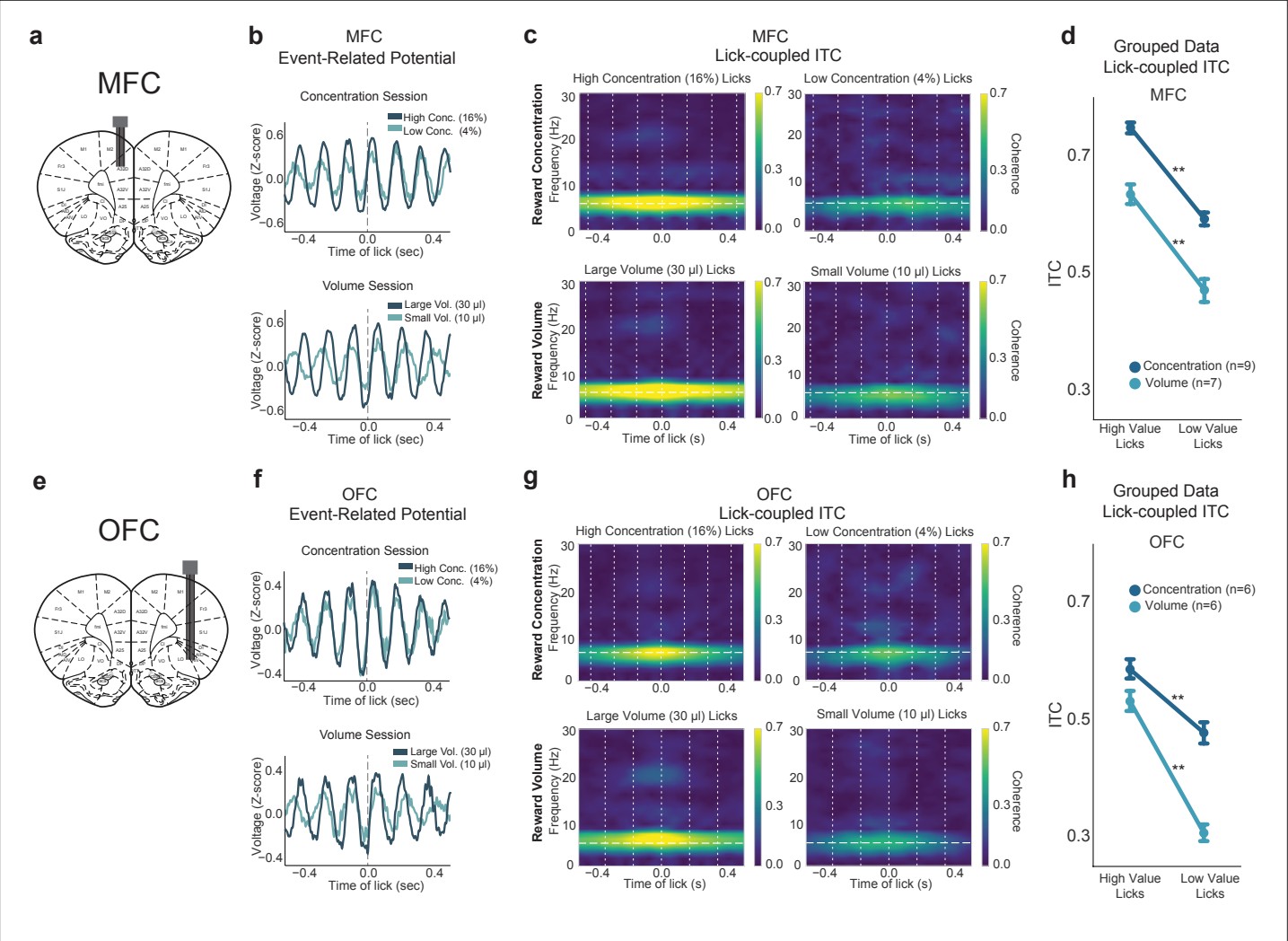

**Figure 4.** Lick-entrained neural activity in medial frontal cortex (MFC) and orbital frontal cortex (OFC) tracked shifts in sucrose concentration and fluid volume. (**a,e**) Rats were implanted with a 2 × 8 electrode array in either MFC (**a**) or OFC (**e**); representative coronal sections are shown. (**b,f**) Event-related potentials during concentration and volume manipulation sessions in the shifting values licking task for MFC (**b**) and OFC (**f**). (**c,g**) Spectral inter-trial coherence (ITC) time-frequency plots revealed strong phase locking during licks for the high concentration and large volume (left sides) rewards in both MFC (**c**) and OFC (**g**). Plots are from one electrode from one individual animal. ITC is consistently strongest around 4–8 Hz. (**d,h**) Grouped data from all rats in both concentration and volume sessions in MFC (**d**) and OFC (**h**) showed strongest ITC during licks for the high-value reward. Double asterisk (**) denotes p < 0.001. Error bars represent 95 % confidence intervals.

The online version of this article includes the following figure supplement(s) for figure 4:

**Figure supplement 1.** Electrophysiological measures used to assess local field potential (LFP) activity.

These findings suggest that LFP activity in both MFC and OFC similarly encodes aspects of preferred versus less preferred reward options; 4–8 Hz phase locking was strongest for both the high concentration and large volume rewards, which may be evidence that the animal is acting within a preferred state with the goal of obtaining their most 'valued' reward. These findings provided further evidence suggesting that the entrainment of neural activity in MFC and OFC to the lick cycle tracks reward magnitude.

## Blocked-interleaved task: engagement in and the vigor of licking vary with reward expectation

The same group of 12 rats were subsequently tested in an adjusted version of the SVLT, which will be referred to as the blocked-interleaved task (*Figure 5a*). In the first 3 min of the task, that is, the

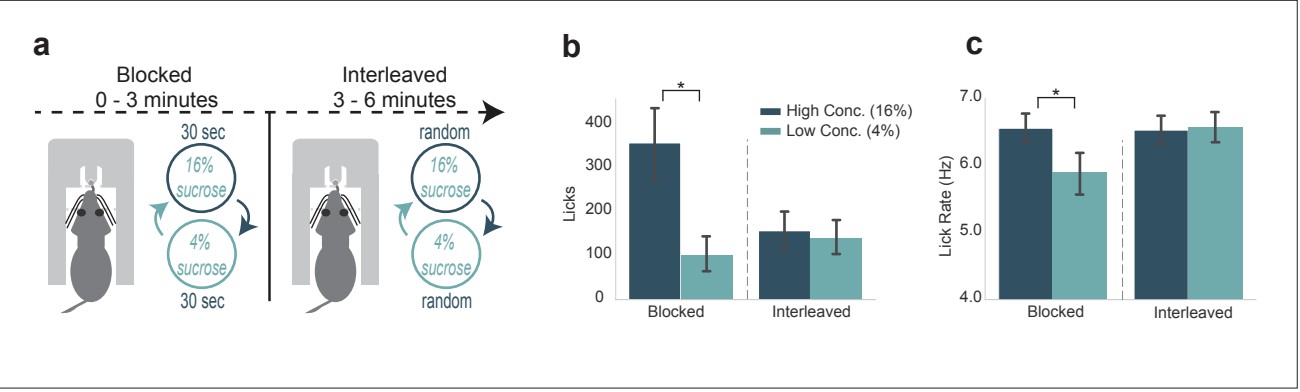

**Figure 5.** Engagement in and the vigor of licking varied with reward expectation. (**a**) Rats participated in a modification of the shifting values licking task, called the blocked-interleaved task, in which they received alternating access to high and low concentrations of liquid sucrose for 3 min and then received interleaved (and thus unpredictable) presentations of the two levels of sucrose for the rest of the session. (**b**) Total licks emitted, a measure of task engagement, for both high and low concentration rewards during the blocked and interleaved portion of the task. Rats licked less for both rewards when rewards were randomly interleaved. (**c**) Lick rate, a measure of response vigor, was similar for both rewards in the interleaved, but not blocked, portion of the task. Asterisk denotes p < 0.05. Error bars represent 95 % confidence intervals.

The online version of this article includes the following figure supplement(s) for figure 5:

**Figure supplement 1.** Transitions in licking behavior in the blocked-interleaved experiment.

'blocked' phase, rats behaviorally showed their typical differentiation of high versus low concentration rewards by emitting more licks for the high concentration reward (**Figure 5b**, left), and licked at a faster rate (**Figure 5c**, left). However, this pattern changed when the rewards were randomly presented in the 'interleaved' part of the task. With a randomly interleaved reward presentation, rats licked nearly equally for high and low concentration rewards (**Figure 5b**, right; see also **Figure 5—figure supplement 1**). We performed a two-way ANOVA on the number of licks by each lick type (high or low concentration) and portion of the task (blocked or interleaved). There was a significant interaction between concentration of reward and the blocked or interleaved portion of the task ($F_{(1,33)}=24.51$, $p < 0.001$). Post hoc analyses revealed that while there was a significant difference in high and low concentration licks during the blocked portion ($p < 0.001$), there was no difference between high and low concentration licks during the interleaved portion of the task ($p = 0.98$). These findings suggest that shifting from blocked to interleaved presentations of the two rewards increased the animals' engagement in licking for the lower value fluid.

Additionally, there was a significant difference in lick rate by each lick type and portion of the task ($F_{(1,33)}=23.13$, $p < 0.001$; two-way ANOVA) (**Figure 5c**). Post hoc analyses revealed that rats licked significantly faster for high versus low concentration rewards during the blocked portion ($p < 0.005$). Lick rates for high versus low concentration licks during the interleaved part of the task were not significantly different ($p = 0.99$). Notably, lick rate during access to either high concentration ($p = 0.005$) or low concentration ($p = 0.002$) rewards during the interleaved portion was significantly increased from lick rate during access to the low concentration reward in the blocked portion of the task. These changes in lick rate were not accounted for by the changes in lick counts reported above (Spearman's rank correlation: 0.44242, $p = 0.20042$) and suggest that shifting from blocked to interleaved presentations of the two rewards increased the vigor with which the rats licked for the lower value fluid.

## Blocked-interleaved task: MFC rhythmicity tracks response vigor

Having established that the blocked-interleaved task can reveal effects of reward expectation on task engagement and response vigor, we next examined how neural activity in the MFC and OFC varies with these behavioral measures. We assessed changes in lick-entrained ERPs and their amplitudes (**Figure 6a and d**), ERSP, and ITC (phase locking) (**Figure 6b,c and e,f**). LFPs in MFC and OFC showed strong 4–8 Hz phase locking during licks for the high concentration rewards in the blocked phase of the task (**Figure 6b and f**). We performed a two-way ANOVA on maximum ITC values (**Figure 6c and e**) from LFPs in both MFC and OFC for each rat and each electrode channel with interaction terms for

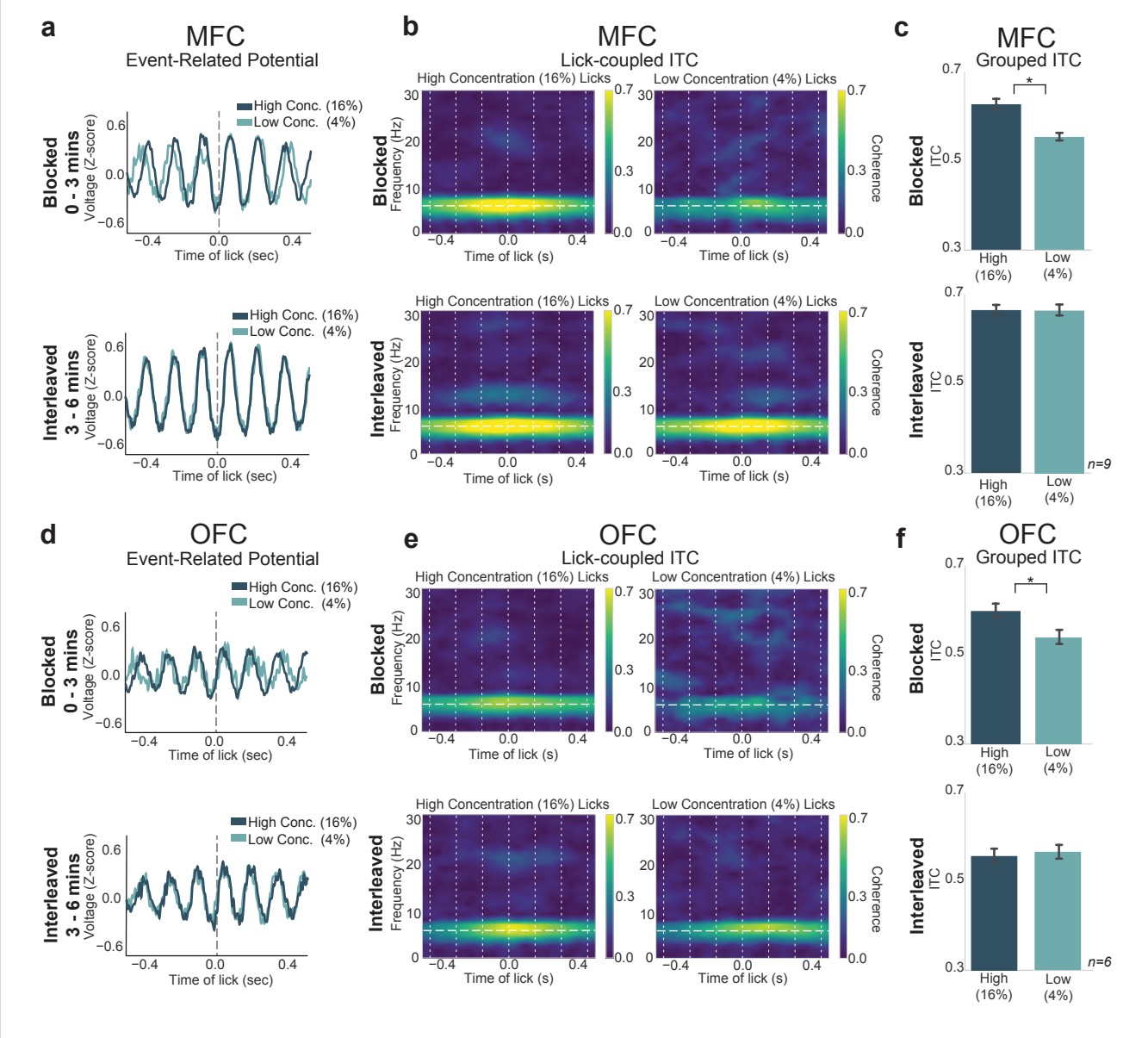

**Figure 6.** Lick-entrained neural activity varied with reward expectation. (**a**,**d**) Event-related potentials (ERPs) for licks of both rewards in medial frontal cortex (MFC) (**a**) and orbital frontal cortex (OFC) (**d**) remain unchanged during the interleaved portion of the task. (**b**,**e**) Spectral inter-trial coherence (ITC) plots revealed stronger 4–8 Hz phase locking during licks for the high concentration reward in the blocked portion (top), but phase locking during licks for high and low concentration rewards in the interleaved portion were indistinguishable from each other. (**c**,**f**) Grouped data revealed no difference in ITC values during high or low concentration licks in the interleaved phase. Asterisk denotes p < 0.05. Error bars represent 95 % confidence intervals.

lick type (high or low concentration reward) and portion of the task (blocked or interleaved reward access), and found a significant interaction of lick type by portion of the task (MFC: $F_{(1,572)}=10.45$, p = 0.001; OFC: $F_{(1,363)}=12.119$, p < 0.001). Post hoc analyses revealed that while there was a significant difference in phase locking of licks for high versus low concentration in the blocked portion (MFC: p < 0.001; OFC: p < 0.036), there was no difference in phase locking of licks for high versus low concentration rewards in the interleaved portion of the task (MFC: p = 0.999; OFC: p = 0.973).

In MFC, a two-way ANOVA revealed a significant interaction of lick type by portion of the task with ERP peak-to-peak size (*Figure 6a*) as the dependent variable ($F_{(1,564)}=6.232$, p = 0.013). However, there were no differences between the ERP measures between high and low concentration licks during the blocked portion of the task (p = 0.887) and between high and low concentration licks during the interleaved portion of the task (p = 0.938).

Neuroscience

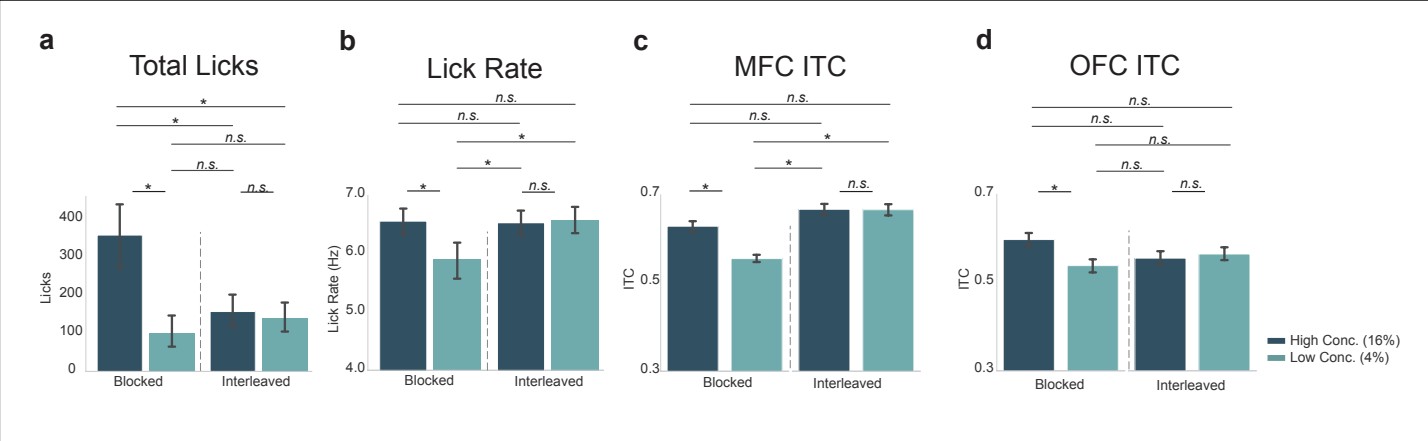

**Figure 7.** Neural activity in medial frontal cortex (MFC), but not orbital frontal cortex (OFC), varied with the lick rate (vigor) and not task engagement (total licks). Post hoc contrasts of statistically significant effects revealed by two-way ANOVA. Direct comparison of behavioral measures (**a** – total licks; **b** – lick rate) with MFC inter-trial coherences (ITCs) (**c**) and OFC ITCs (**d**) showed a similar pattern (and identical post hoc statistical contrasts) between lick rate (**b**) and MFC ITCs (**c**). The pattern of post hoc contrasts for OFC ITCs (**d**) did not match either total licks or lick rate. Asterisk denotes p < 0.05. Error bars represent 95 % confidence intervals.

The same was true with ERSP measures for MFC LFPs. There was a significant interaction between lick type and portion of the task (F(1,564)=30.17, p < 0.001; two-way ANOVA), but no significant difference between ERSP values between high and low concentration licks in the blocked (p = 0.213) or interleaved (p = 0.743) portions of the task. In OFC (*Figure 6d*), there was no significant interaction of lick type and portion of the task by the amplitude size of the lick's ERPs (F(1,363)=0.131, p = 0.718; two-way ANOVA), and no difference in OFC ERSP values of lick type by portion of the task either (F(1,363)=0.744, p = 0.389; two-way ANOVA).

We wanted to further investigate potential differences in MFC and OFC in the blocked-interleaved task, since initial results show a general increase of ITC values from MFC in the interleaved portion of the task and a general decrease in ITC values from OFC. This was of particular interest since MFC ITC values varied with the lick rate, which increased for both the high and low concentration licks in the interleaved portion of the task. We directly compared ITC values in both regions with lick-rate and total-lick counts (*Figure 7*).

Post hoc analyses displayed in *Figure 7c* revealed that in MFC there was a significant difference between ITC values for the high versus low concentration licks (as also documented at the top of *Figure 6c*), but ITC values for high concentration licks during the blocked portion of the task did not differ from ITC values for either the high (p = 0.075) or low concentration (p = 0.089) conditions in the interleaved portion of the task. The pattern of post hoc contrasts matches the lick-rate data (*Figure 7b*) for all paired comparisons. This match includes the finding (*Figure 7c*) that ITC values for low concentration licks in MFC differed from all three of the other conditions (high concentration blocked, high concentration interleaved, and low concentration interleaved licks; p < 0.001 for each comparison). The MFC ITC post hoc test results (*Figure 7c*) did not match the pattern for total licks (*Figure 7a*).

In OFC, ITC values (*Figure 7d*) did not match either the total-lick (*Figure 7a*) or lick-rate (*Figure 7b*) comparisons, despite the qualitative similarity with the total number of licks (compare *Figure 7d* with *Figure 7a*). The only significant difference in ITC values in OFC was between the high and low concentration licks in the blocked portion of the task (as also documented at the top of *Figure 6f*). All other comparisons were non-significant. This pattern of post hoc comparisons did not match either total licks (compare *Figure 5a with d*) or lick rate (compare *Figure 7b with d*).

Together with the results summarized in *Figure 6*, these findings from post hoc testing in *Figure 7* provide evidence that MFC and OFC encode different aspects of licking and reward value. There was a clear match between the pattern of lick entrainment in the MFC, but not the OFC, with the animals' licking rates. The correspondence between lick entrainment in MFC and the animals' lick rates provides support for the idea that neural activity in MFC is sensitive to response vigor. By contrast,

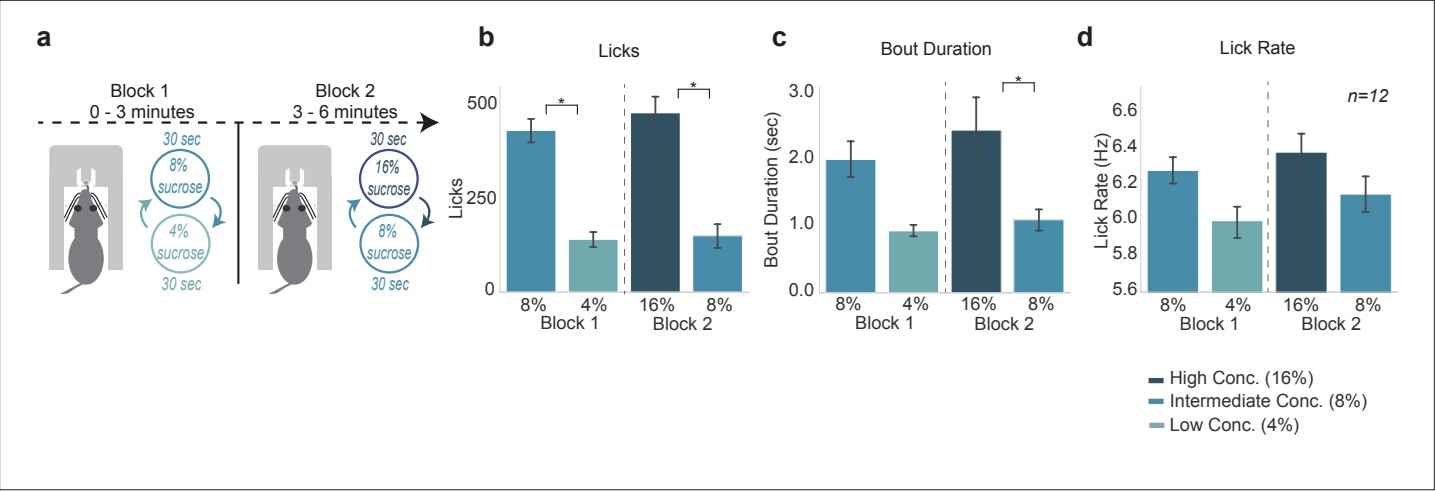

**Figure 8.** Consummatory behavior tracked relative differences in reward value. (**a**) The three reward task is a variation of shifting values licking task but with a third reward introduced. In the first block of the task, rats experience the intermediate (8%) reward and low (4%) reward. In block 2, rats experience the high (16%) reward paired with the intermediate (8%) reward. (**b**) Rats licked more for the sweeter reward in each block. (**c**) Rats showed greater bout durations for the sweeter reward. (**d**) Lick rate showed a similar pattern to licks and bout duration, but was not statistically significant. Asterisk denotes p < 0.05. Error bars represent 95 % confidence intervals.

OFC might be involved in more general aspects of motivation, for example, to lick or not (reward evaluation) based on reward magnitude or the predictability of the environment.

## Three reward task: behavioral evidence for effects of relative reward value

The previous experiments assessed comparison of two levels of rewards (either high/low concentration or large/small volume) in the SVLT. After finding behavioral and electrophysiological differences between two rewards, we aimed to investigate how animals process reward with contexts involving three different rewards. In this experiment, we assessed if rats process rewards in a relative manner or in an absolute manner by implementing a third intermediate (8% wt/vol sucrose concentration) reward.

In the three reward task (*Figure 8A*), the first block consists of the SVLT with 30 s shifts between the intermediate-value (8 % sucrose) reward and the low-value (4 % sucrose) reward. After 3 min the second block of the task begins, where rats then experience shifting values of reward from the high-value (16 % sucrose) reward to the intermediate-value (8 % sucrose) reward. This allowed us to assess how rats would process the intermediate 8 % sucrose reward when it is paired with a worse (4%) or better (16%) option within one session. Additionally, the design introduces a second context (just like in the blocked-interleaved task previously) in which we could assess if animals are still processing a (temporally) local comparison of reward types.

Licking varied with both reward value and block, that is, low versus intermediate and intermediate versus high (F(3,33)=34.2, p < 0.001) (*Figure 8b*). Post hoc analyses revealed that rats emitted significantly more licks for the intermediate value 8 % reward as opposed to the low value 4 % reward in block 1 (p < 0.001). In block 2, rats also emitted significantly fewer licks for the intermediate value 8 % reward when it was paired with the high value 16 % reward (p < 0.001). Rats also licked significantly less for the intermediate 8 % reward in block 2 than they did in block 1 (p < 0.001).

There was a more subtle effect for differences in bout duration across the different rewards (F(3,33)=5.333, p = 0.004; two-way ANOVA) (*Figure 8c*). Post hoc analyses revealed no significant difference in bout duration for the 4 % versus 8 % in block 1 (p = 0.098), yet there was a significant decrease in bout durations during access to the 8 % versus 16 % in block 2 (p = 0.023). Bout durations during access to the intermediate 8 % reward in block 1 versus block 2 were not different (p = 0.20). While there was a significant effect of lick type on lick rate (F(3,33)=10.59, p < 0.001; two-way ANOVA), post hoc analyses revealed no major differences in lick rate of the licks for rewards in block

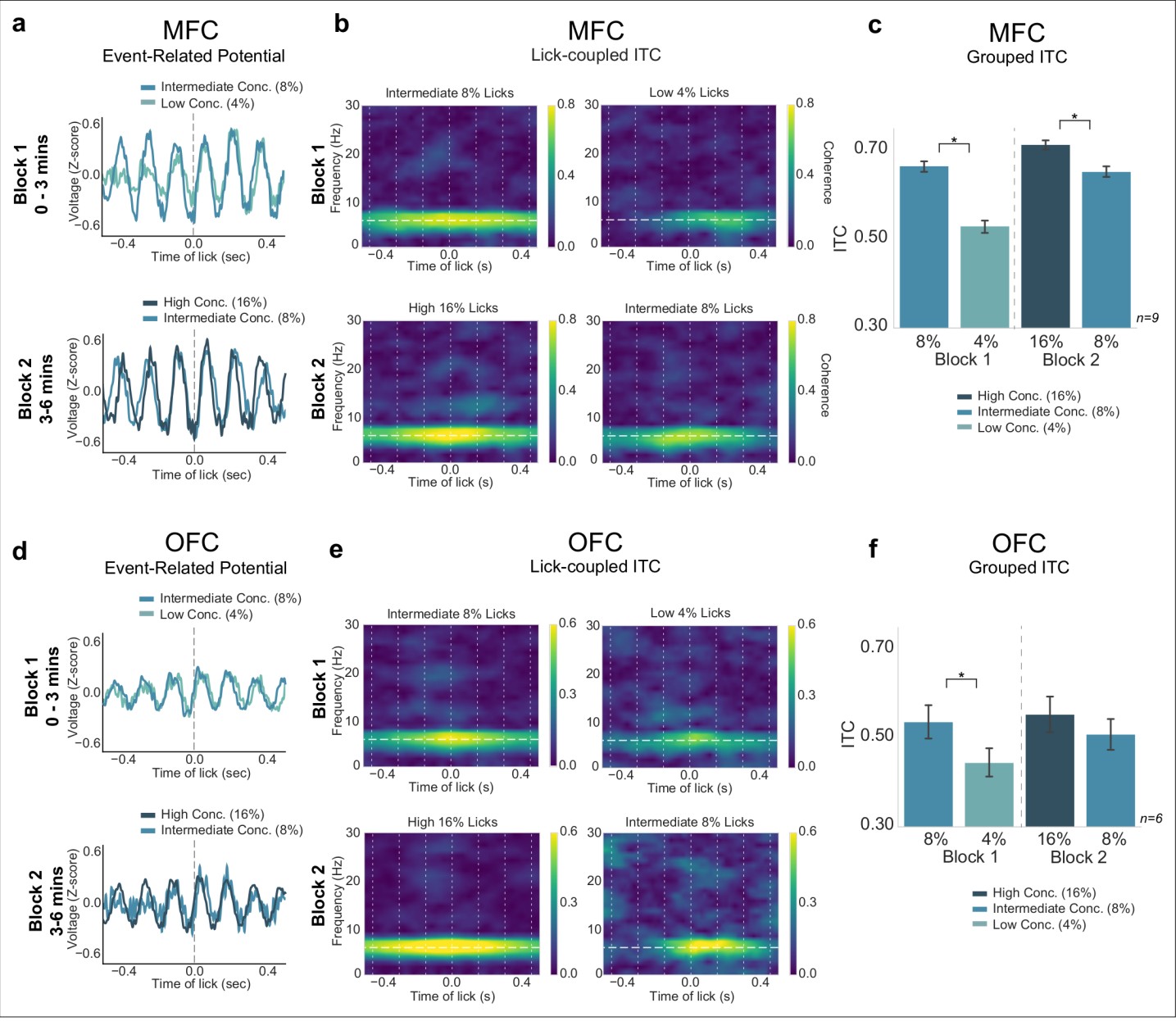

**Figure 9.** Neural activity in medial frontal cortex (MFC), but not orbital frontal cortex (OFC), tracked absolute differences in reward value. (**a**,**d**) Event-related potentials (ERPs) for each block of the task from MFC (**a**) and OFC (**d**). (**b**,**e**) Inter-trial coherence (ITC) values in MFC (**b**) and OFC (**e**) showed strongest 4–8 Hz phase locking for the 'high-value' reward in each block. (**c**,**f**) Group data revealed significantly greater ITC values for the high-value reward in each block for MFC ITCs (**c**), and a similar pattern was found in OFC (**f**) but only block 1 rewards were significantly different. Asterisk denotes p < 0.05. Error bars represent 95 % confidence intervals.

1 (p = 0.17) or block 2 (p = 0.31) (*Figure 8d*), nor for the lick rate for 8 % licks in block 1 versus block 2 (p = 0.76).

## Three reward task: neural activity does not reflect relative reward value encoding

The behavioral measures summarized above established that the three reward task can reveal effects of relative value comparisons. We next analyzed electrophysiological signals from MFC and OFC (*Figure 9*) to determine if they tracked the animals' behavior in the task, and might encode relative differences in value, or some other aspect of value, such as the absolute differences between the three rewards. We found a significant difference between ITC values for the three different rewards in both

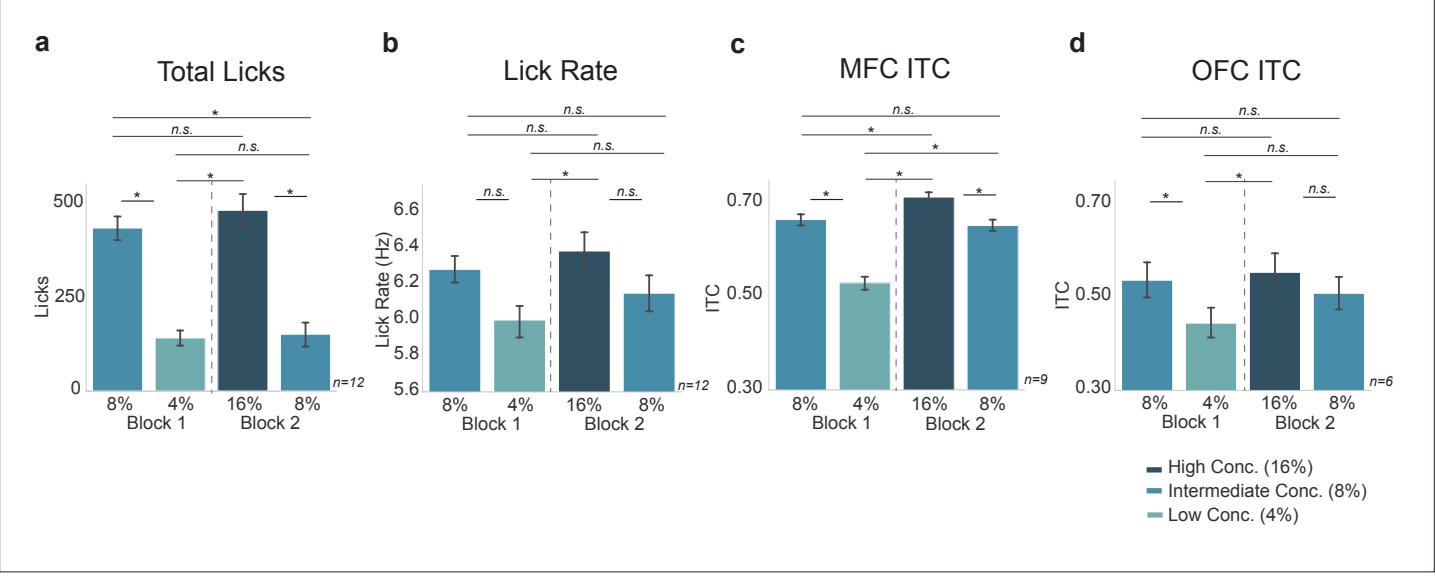

**Figure 10.** Neural activity in medial frontal cortex (MFC), but not orbital frontal cortex (OFC), varied with effects of absolute reward value on lick rate (vigor) and task engagement (total licks). (**a**,**b**) Behavioral measures replotted with significance bars for each combination reward. MFC inter-trial coherences (ITCs) (**c**) did not show the exact same pattern as lick rate, which is different from *Figure 5*. OFC ITCs (**d**) did not look like total licks or lick rate. Asterisk denotes p < 0.05. Error bars represent 95 % confidence intervals.

The online version of this article includes the following figure supplement(s) for figure 10:

**Figure supplement 1.** Hypothesis for relative versus absolute encoding of reward value.

**Figure supplement 2.** Third block of testing in the three reward task.

MFC and OFC (MFC: F(3,627)=154.4, p < 0.001; OFC: F(3,363)=13.29, p < 0.001; two-way ANOVAs). Tukey post hoc analyses revealed a difference in ITC values between intermediate and low licks in block 1 (MFC: p < 0.001; OFC: p = 0.003), and a difference in ITCs between high and intermediate licks in block 2 for MFC only (MFC: p < 0.005; OFC: p = 0.313)(*Figure 9b, c and e,f*). There was no difference between ITC values from intermediate (8%) block 1 and intermediate block 2 licks in both regions (MFC: p = 0.881; OFC: p = 0.705). There was a significant difference between MFC ITC values for block 1 intermediate (8%) licks and block 2 high (16%) licks (p = 0.028), as well as a significant difference between MFC ITC values for block 1 low (4%) licks and block 2 intermediate (8%) licks (p < 0.001). Signals from the OFC did not differ across these conditions.

Peak-to-peak amplitude analysis of the three reward task revealed a significant effect of block on MFC LFP amplitude across lick types (F(3,627)=15.56, p < 0.001; two-way ANOVA) (*Figure 9a*). Tukey post hoc testing revealed no relevant significant differences between ERP size in MFC (between block 1 intermediate and low licks: p = 0.864; between block 2 high and intermediate licks: p = 0.944). There was no difference in OFC amplitude size (F(3,363)=0.827, p = 0.479, two-way ANOVA) (*Figure 9d*). While there was a significant effect for ERSP values in both MFC and OFC (MFC: F(3,627)=18.35, p < 0.001; OFC: F(3,363)=5.108, p = 0.002; two-way ANOVAs), none of the relevant measures were significant (block 1 intermediate and low licks: MFC: p = 0.875; OFC: p = 0.492; block 2 high and intermediate licks: MFC: p = 0.637; OFC: p = 0.999).

The ITC findings, at least in MFC, support the idea that the 'higher value' and 'lower value' rewards in each context are being encoded differently across contexts. They indicate that MFC might instead encode absolute reward value instead of relative reward value. Qualitatively, the ITC values in MFC seem to have the same pattern as the lick rate (*Figure 10*), similar to how MFC values reflected lick rate in the blocked-interleaved task. However, post hoc statistical testing revealed important differences. For example, the ITC in MFC differed significantly for high- versus low-value rewards in both blocks 1 and 2, but lick rate did not. Importantly, post hoc analyses revealed a significant difference in ITC values in MFC for every reward combination except for the intermediate block 1 and intermediate block 2 rewards, which reflects our operational definition for absolute encoding of value (see *Figure 10—figure supplement 1a, b*).

The encoding of value was less clear based on ITC measures from the OFC. These values did not directly match the licking behavior (in either rate, total licks, or bout duration) (compare *Figure 10* with *Figure 8d*), and did not show clear evidence for either absolute or relative encoding of reward. Instead, the results from *Figure 10D* indicate that OFC might instead encode reward value in a mixed absolute/relative manner (as in *Figure 10—figure supplement 1* and *Figure 10—figure supplement 2*). However, these findings should be interpreted in the light of uneven sampling between areas, with fewer recordings done in the OFC. It is therefore possible that our results are underpowered for the OFC and new experiments could reveal an alternative interpretation.

## Discussion

We investigated the role of MFC and OFC in processing reward information as rats participated in various consummatory licking tasks. Rats process and express changes in reward size in roughly the same manner as with reward concentration, both behaviorally and electrophysiologically. LFP activity in both MFC and OFC is sensitive to changes in reward type (both volume and concentration). Our results reveal context-dependent value signals in both regions through randomly presented rewards and by introducing a third reward in the task. Behaviorally, rats show evidence for a relative expression of rewards, while neural activity in MFC and OFC did not reflect relative encoding of reward. Together, our findings suggest that rats sample rewards and commit to consuming a given reward when they are able to predict its value, and this behavior is coupled to neural activity in MFC and OFC that encode both the value of the reward and the animal's consummatory strategy. The subtle differences between the two regions follow the hypothesis that these areas provide different roles during consummatory behavior. We additionally provide evidence for MFC representing action-outcome relationships, as MFC ITC activity is more strongly correlated to the action of licking and may signal information about the 'value of the action'.

### Rhythmic activity and reward processing

Similar to our previous studies (*Horst and Laubach, 2013*; *Amarante et al., 2017*), neural activity was entrained to the lick cycle across all tasks in both MFC and OFC. Entrainment was strongest for the high-value reward (either of size or sweetness) and varied with the animals consummatory strategy (persistently lick a highly preferred option or sample fluid and wait for better option). Previous studies have viewed this rhythmic activity as being driven by the act of licking, as rats naturally lick at 6–7 Hz (*Travers et al., 1997*; *Weijnen, 1998*; *Horst and Laubach, 2013*). However, the activity cannot be explained solely by licking, as there are instances where phase locking and behavior do not show the same pattern (e.g. the blocked-interleaved experiment), and the variety of studies reported here and in *Amarante et al., 2017*, suggest a higher order role for rhythmic activity in the control of consummatory behavior.

Indeed, a major question for our study might be the extent to which lick-entrained oscillations in MFC and OFC can be dissociated from the act of licking. We adapted methods for spike-field coherence to examine this question (*Figure 2*). We found that the coherence between licks and rhythmic LFP signals in the licking frequency range was no larger than 0.5 (with lick-field coherence ranging between 0 and 1), was stronger for the MFC recordings compared to the OFC recordings, and varied with the reward value of the consumed fluid. These findings suggest that the LFP oscillations are not simply driven by licking.

Furthermore, using directed coherence to examine directional influences of recordings in the two cortical areas, we found evidence for a weak directionality at the licking frequency such that the phase of the signals in MFC lead those in OFC. This result was especially apparent for the most rostral recording sites in the MFC, located in the medial orbital area and the adjacent frontal agranular area. Notably, these recording sites are immediately adjacent to a region of the frontal cortex where oral movements can be generated by electrical stimulation at low current (*Yoshida et al., 2009*). As such, the field recordings in the rostral part of the MFC might reflect activity from the adjacent oral motor cortex or could be locally generated. Resolving this matter will require new experiments, likely using optogenetic stimulation to avoid stimulation of fibers of passage.

We propose a functional interpretation of these signals based on findings on 'medial frontal theta' (*Cavanagh and Frank, 2014*) in other types of behavioral tasks. There have been several proposals

for the role of frontal theta in information processing. One idea is that the rhythm acts to break up sensory information into temporal chunks (*Uchida and Mainen, 2003*), and is related to the notion of a global oscillatory signal to synchronize neural activity across multiple brain structures throughout the taste-reward circuit (*Gutierrez and Simon, 2013*). Another idea is that frontal theta acts as an action monitoring signal (*Cavanagh et al., 2012*; *Narayanan et al., 2013*; *Laubach et al., 2015*), which can be generated through simple recurrent spiking network models (*Bekolay et al., 2014*). Finally, instead representing a specific function, frontal theta may act as a convenient 'language' for distant brain regions to exchange information with each other (*Womelsdorf et al., 2010*). Our general findings contribute to this literature by suggesting that frontal theta acts as a value signal to guide consummatory behavior, which is the ultimate consequence of many goal-directed actions in natural environments.

## A common code for reward magnitude

A major finding in the present study (*Figures 3 and 4*) was the similar electrophysiological signals in MFC and OFC are associated with the consumption of high and low concentration liquid sucrose rewards and large and small volume rewards. Although other studies have found either decreases (*Kaplan et al., 2001*) or increases in behavior with increases in concentration and volume rewards in the same study (*Hulse et al., 1960*; *Collier and Myers, 1961*; *Collier and Willis, 1961*), these studies did not investigate the electrophysiological correlates of consuming rewards. Our study is the first to show a generalized 'value signal' in the frontal cortex that scales with increased size and increased concentration of liquid sucrose. These signals might underlie the computation of a common currency (*Montague and Berns, 2002*; *Levy and Glimcher, 2011*; *Levy and Glimcher, 2012*; *Strait et al., 2014*) for the amount of nutrient available in a given food item and contribute to value-guided control of consumption.

## Evidence for the contextual control of consumption

In the blocked-interleaved task (*Figure 5A*), rats who licked more, longer, and faster for the high concentration reward when rewards were blocked did not continue to do so during interleaved portion of the task (*Figure 5B–C*). Instead, they licked nearly equally for the high and low concentration solutions, a result that is suggestive of the loss of positive contrast effects for the higher value fluid that is commonly found in the blocked design (*Parent et al., 2015a*). Despite these differences in behavior, the rats' LFPs in MFC and OFC showed high levels of lick-entrained activity, essentially equal to that found during consumption of the higher value fluid in the blocked part of the session.

This finding is hard to reconcile with enhanced lick entrainment reflecting reward contrast effects. If positive contrast engenders entrainment, then LFPs should have shown reduced phase locking to the lick cycle in the interleaved portion of the task. Instead, the results might suggest that LFPs in MFC and OFC are entrained to licking when rats engage in persistent licking, as was found in the periods with high concentration access in the blocked part of the sessions and across the entire interleaved part of the session, and entrainment is reduced when rats switch to sampling the fluid during periods with low-value access in the blocked part of the session. By this view, LFP entrainment to the lick cycle could serve as a contextual marker for reward state and the behavioral strategy deployed by the rat to sample and wait or persistently consume the liquid sucrose. This contextual information would depend on knowledge of the temporal structure of the reward deliveries. That is, when reward values are blocked, the rats have learned to expect alternative access to higher and lower reward values over extended periods of time (30 s). By contrast, when reward values are interleaved, the changes in values occur rapidly and are unpredictable. The reduction in lick entrainment might therefore reflect the animal's sampling strategy.

Contextual coding of reward value was also apparent in the three reward task (*Figures 8–10*), where lick entrainment was stronger when the higher value option was available (*Figure 9*). In this case, the strength of engagement, for MFC but not OFC, tracked reward value in manner suggestive of an absolute reward encoding, with entrainment being higher for the 16 % sucrose solution compared to the 8 % solution when both were the 'best' option (*Figure 10C and D*). These electrophysiological results were notably distinct from behavioral measures such as total licking output and lick rate (*Figure 10A and B*), which provided evidence for relative value comparisons.

Our electrophysiological results support theories of absolute reward value (*Hull, 1943*; *Spence, 1956*; *Flaherty, 1982*), as opposed to theories of relative reward value (*Crespi, 1942*; *Black, 1968*; *Webber et al., 2015*). Our findings might also fit with the neuro-economics idea of menu invariance versus menu-dependent goods (*Padoa-Schioppa, 2011*), both of which have been supported by electrophysiological studies on OFC (*Padoa-Schioppa and Assad, 2006*; *Padoa-Schioppa and Assad, 2008*; *Tremblay and Schultz, 1999*; *Saez et al., 2017*).

Notably, in several instances we found a mismatch of behavioral output and corresponding magnitude of neural activity. This was evident in the blocked-interleaved task, where MFC and OFC ITCs did not reflect total licks emitted, as well as in the three reward task where MFC and OFC ITCs did not reflect total licks or lick rate. This is in opposition to the SVLT, where MFC and OFC activity directly matched behavioral output of licks, lick rate, and bout duration. These findings reveal the importance of recording careful behavioral output with electrophysiological recordings, and it remains an open discussion on the mechanisms behind correlative behavior versus diverging behavioral output from neural activity.

## Functional interpretations of phase entrainment

The original observation that suggested phase locking of licks to MFC field potentials was reported in *Horst and Laubach, 2013*. Peri-event plots of LFPs around the times of licks revealed ERPs. The nature of ERPs has been researched extensively in the EEG literature. A leading view is that ERPs arise from a synchronization of the phase of an ongoing rhythm and/or from the superposition of inputs to the region of cortex near the electrode (e.g. *Klimesch et al., 2007*; *Sauseng et al., 2007*). Evidence for phase locking near the licking frequency can be found in Figures 1 and 3 in *Amarante et al., 2017*, with some exceptions being at slightly higher frequencies, for example, *Figure 8E* in that study. By contrast, LFP power is typically tonic in the range of delta (<4 Hz) and the animals' licking frequency mostly showed only minor changes in power (Figure 1 and 3 in *Amarante et al., 2017*). Furthermore, in another experiment with periodic reinforcement, we reported that phase but not power varied reinforcement (*Figure 7*, *Amarante et al., 2017*). These findings suggest that phase, not power, has a relationship with reinforced licking behavior, and the same determinants for phase locking likely apply to the results reported here.

Our data suggest that the act of licking synchronizes the phase of ongoing rhythms in the MFC and OFC and that this synchronization occurs during periods of sustained increases in delta band power. Computational models of LFP rhythmicity suggest that information flow is controlled by the interplay between different functional rhythms, with activity at higher frequencies nested within the periods of lower frequencies (*Kopell et al., 2010*). Brain slice (*Carracedo et al., 2013*) suggests that theta-range rhythmicity may be nested within cycles of the lower frequency delta rhythm. For our studies, as shown in Figure 3 from *Amarante et al., 2017*, the duration of elevated phase synchronization was roughly twice as long as the median inter-lick interval, and the inverse of this interval would indicate a frequency of ~3.5 Hz, that is, delta. The same mechanisms are likely to apply to the present study. A possible source for the delta rhythm is the animal's respiratory cycle (*Lockmann and Tort, 2018*), which must be regulated during periods of sustained licking when high-value fluid is available.

Our finding of zero-lag correlation across frequencies (*Figure 2D*) further suggests that a source of common variance to MFC and OFC modulates the timing of processing, and leads to a slight advance in the phase angles of the rhythm in MFC relative to OFC (directed coherence in *Figure 2E–H*). The strength of this input would seem to vary between areas, being stronger in MFC compared to OFC and strongest in the rostral MFC (medial orbital cortex) overall (*Figure 2H*). These patterns of directional influences presumably vary with the extent of lick-field coherence and thereby with the value of the consumed fluid.

It is not clear from our studies if the reduction in entrainment when low-value rewards are available is an active or passive process. For example, it is possible that some active input to the MFC and OFC denotes the temporal context (e.g. dopamine, hippocampus), enhancing entrainment when the higher value option is available. Alternatively, signals from sensorimotor regions of the frontal cortex, which sit in between the MFC and OFC, the oral sensory and motor cortices (*Yoshida et al., 2009*), might be reduced during periods with less intense licking, leading to a passive reduction in overall frontal lick entrainment. Future studies are needed to address these neural mechanisms of licking-related synchrony in the rodent frontal cortex.

## Differences in reward signaling between MFC and OFC

The electrophysiological results from the blocked-interleaved task and three reward task suggest that MFC and OFC, while showing similar results overall, may be contributing to processing reward information in different ways. It is important to note that due to a smaller sample size of OFC recordings, the less clear findings in OFC may indeed require further future experiments. However, our findings do follow previous work on subtle differences of these areas. In accord with a previous theory on proposed MFC and OFC functions (*Balleine and Dickinson, 1998*; *Balleine and Dickinson, 2000*; *Schoenbaum et al., 2009*; *Sul et al., 2011*; *Passingham and Wise, 2012*), MFC activity may be acting to maintain and optimize licking behavior in an action-centric manner, as reflected in measures such as the licking rate, a measure associated with vigor and sensitive to inactivation of the same cortical area in a progressive ratio licking task (*Swanson et al., 2019*). By contrast, OFC activity generally reflected differences in reward value, perhaps due to the different sensory properties of the fluids (*Gutierrez et al., 2006*), and was not sensitive to licking rate (vigor) or task engagement (total licks).

# Materials and methods

**Key resources table**

| Reagent type (species) or resource | Designation | Source or reference | Identifiers | Additional information |
|---|---|---|---|---|
| Strain, strain background (*Rattus norvegicus*) | Sprague-Dawley, Long-Evans | Charles River, Envigo | NA | Rat (male) |
| Other | Precision Syringe Pump Controller | https://doi.org/10.1523/ENEURO.0240-19.2019 | SCR_021493 | |
| Software, algorithm | Med-PC | MedAssociates | SCR_012156 | |
| Software, algorithm | GNU Octave | https://www.gnu.org/software/octave/ | SCR_014398 | |
| Software, algorithm | R Project for Statistical Computing | https://www.r-project.org/ | SCR_001905 | |
| Software, algorithm | NeuroExplorer | https://www.neuroexplorer.com/ | SCR_001818 | |
| Software, algorithm | MatPlotLib | https://matplotlib.org/ | SCR_008624 | |
| Software, algorithm | IPython | https://ipython.org/ | SCR_001658 | |
| Software, algorithm | Jupyter | https://jupyter.org/ | SCR_018416 | |
| Software, algorithm | Seaborn | https://seaborn.pydata.org/ | SCR_018132 | |
| Software, algorithm | MATLAB | Mathworks | SCR_001622 | |
| Software, algorithm | EEGLab | https://sccn.ucsd.edu/eeglab/index.php | SCR_007292 | |

All procedures carried out in this set of experiments were approved by the Animal Care and Use Committee at American University (Washington, DC). Procedures conformed to the standards of the National Institutes of Health Guide for the Care and Use of Laboratory Animals. All efforts were taken to minimize the number of animals used and to reduce pain and suffering.

## Animals

Male Long-Evans and Sprague-Dawley rats weighing between 300 and 325 g were used in these studies (Charles River, Envigo). As relatively few animals were used, we did not investigate sex differences in reward processing in this study. Sex differences among rats are well known for how liquid sucrose is consumed (e.g. *Sclafani et al., 1987*) and classic studies of incentive contrast (e.g. *Flaherty and Rowan, 1986*), which led to the design of the behavioral procedures used here, were mostly carried out using male rats. As such, we cannot comment on sex differences or how reward value is encoded in the frontal cortex of female rats. These important topics require further study.

Rats were given 1 week to acclimate with daily handling prior to behavioral training and surgery and were then kept with regulated access to food to maintain 90 % of their free-feeding body weight. They were given ~18 g of standard rat chow each day in the evenings following experiments. Rats were single housed in their home cages in a 12 hr light/dark cycle colony room, with experiments

occurring during the light cycle. A total of 12 rats had a 2 × 8 microwire array implanted into either the MFC (N = 6), the OFC (N = 2), or one array in each area contralaterally (N = 4). Arrays consisted of 16 blunt-cut 50 μm tungsten (Tucker-Davis Technologies) or stainless steel (Microprobes) wires, separated by 250 μm within each row and 500 μm between rows. In vitro impedances for the microwires were ~150 kΩ.

## Surgeries

Animals had full access to food and water in the days prior to surgery. Stereotaxic surgery was performed using standard methods. Briefly, animals were lightly anesthetized with isoflurane (2.5 % for ~2 min), and were then injected intraperitoneally with ketamine (100 mg/kg) and dexdomitor (0.25 mg/kg) to maintain a surgical plane of anesthesia. The skull was exposed, and craniotomies were made above the implant locations. Microwire arrays were lowered into MFC (coordinates from bregma AP: +3.2 mm; ML: +1.0 mm; DV: –1.2 mm from the surface of the brain, at a 12-degree posterior angle; *Paxinos and Watson, 2013*) or into OFC (AP: +3.2 mm, ML: +4.0 mm, DV: –4.0 mm; *Paxinos and Watson, 2013*). The part of the MFC studied here is also called 'medial prefrontal cortex' in many rodent studies and the region is thought to be homologous to the rostral ACC of primates (*Laubach et al., 2018*). Four skull screws were placed along the edges of the skull and a ground wire was secured in the intracranial space above the posterior cerebral cortex. Electrode arrays were connected to a head-stage cable and modified Plexon preamplifier during surgery, and recordings were made to assess neural activity during array placement. Craniotomies were sealed using cyanocrylate (Slo-Zap) and an accelerator (Zip Kicker), and methyl methacrylate dental cement (AM Systems) was applied and affixed to the skull via the skull screws. Animals were given a reversal agent for dexdomitor (Antisedan, 0.25 mg/mL, s.c.), and Carprofen (5 mg/kg, s.c.) was administered for postoperative analgesia. Animals recovered from surgery in their home cages for at least 1 week with full food and water, and were weighed and monitored daily for 1 week after surgery.

## Behavioral apparatus

Rats were trained in operant chambers housed within a sound-attenuating external chamber (Med Associates; St. Albans, VT). Operant chambers contained a custom-made glass drinking spout that was connected to multiple fluid lines allowing for multiple fluids to be consumed at the same location. The spout was centered on one side of the operant chamber wall at a height of 6.5 cm from the chamber floor. Tygon tubing connected to the back of the drinking spout administered the fluid from a 60 cc syringe hooked up to either a PHM-100 pump (Med Associates) for standard experiments or to a customized open-source syringe pump controller (*Amarante et al., 2019*) that is programmed by a teensy microcontroller to deliver different volumes of fluid with the same delivery time from one central syringe pump. A 'light-pipe' lickometer (Med Associates) detected licks via an LED photobeam, and each lick triggered the pump to deliver roughly 30 μL per 0.5 s. Behavioral protocols were run though Med-PC version IV (Med Associates), and behavioral data was sent via TTL pulses from the Med-PC software to the Plexon recording system.

## Shifting values licking task

The operant licking task used here is similar to those previously described (*Parent et al., 2015a*; *Parent et al., 2015b*; *Amarante et al., 2017*). Briefly, rats were placed in the operant chamber for 30 min, where they were solely required to lick at the drinking spout to obtain a liquid sucrose reward. Licks to the light-pipe lickometer would trigger the syringe pump to deliver liquid sucrose over 0.5 s. In other words, the first lick to the spout triggers the pump and reward is then delivered for 0.5 s, where any lick within that 0.5 s window would also be rewarded. The next lick after 0.5 s would subsequently trigger the pump to turn on again for 0.5 s. Every 30 s, the reward alternated between high (16% wt/vol) and low (4% wt/vol) concentrations of liquid sucrose, delivered in a volume of 30 μL. In volume manipulation sessions, the reward alternated between a large (27.85 μL) and small volume (9.28 μL) of 16 % liquid sucrose. Rewards were delivered over a period of 0.5 s for all levels of concentration and volume using a custom-made syringe pump (*Amarante et al., 2019*). The animal's licking behavior was constantly recorded throughout the test sessions.

### Blocked versus randomly interleaved licking task

The SVLT was altered to allow for comparison of blocked versus interleaved presentations of reward values. The first 3 min of the task consisted of the standard SVLT, with 30 s blocks of either the high or low concentration sucrose rewards delivered exclusively during the block. After 3 min, the rewards were presented in a pseudo-random order (e.g. high, high, low, high, low, low, high) for the rest of the test session. With rewards interleaved, rats were unaware of which reward would be delivered next. Behavioral and neural data were only analyzed from the first 6 min of each test session. We focused on manipulating sucrose concentration, and not fluid volume, in this task variation, as concentration differences provided the most effects of reward value on licking behavior (see *Figure 1D*).

### Three reward licking task

The SVLT was modified, using a third intermediate concentration of sucrose (8% wt/vol) to assess if reward value influenced behavior and neuronal activity in a relative or absolute manner. In the first 3 min of each session, rats received either the intermediate (8%) or low (4%) concentration of sucrose, with the two rewards delivered over alternating 30 s periods as in the SVLT. After 3 min, the rewards switched to the high (16%) and intermediate (8%) concentrations, and alternated between those concentrations for the rest of the session. Behavioral and neural data were only analyzed from the first 6 min of each test session.

### Electrophysiological recordings

Electrophysiological recordings were made using a Plexon Multichannel Acquisition Processor (MAP; Plexon; Dallas, TX). LFPs were sampled on all electrodes and recorded continuously throughout the behavioral testing sessions using the Plexon system via National Instruments A/D card (PCI-DIO-32HS). The sampling rate was 1 kHz. The head-stage filters (Plexon) were at 0.5 Hz and 5.9 kHz. Electrodes with unstable signals or prominent peaks at 60 Hz in plots of power spectral density were excluded from quantitative analysis.

### Histology

After all experiments were completed, rats were deeply anesthetized via an intraperitoneal injection of Euthasol (100 mg/kg) and then transcardially perfused using 4 % paraformaldehyde in phosphate-buffered saline. Brains were cryoprotected with a 20 % sucrose and 10 % glycerol mixture and then sectioned horizontally on a freezing microtome. The slices were mounted on gelatin-subbed slides and stained for Nissl substance with thionin.

### Data analysis: software and statistics

All data were analyzed using GNU Octave (https://www.gnu.org/software/octave/), Python (Anaconda distribution: https://www.continuum.io/), and R (https://www.r-project.org/). Analyses were run as Jupyter notebooks (http://jupyter.org/). Computer code used in this study is available upon request from the corresponding author.

Statistical testing was performed in R. Paired t-tests were used throughout the study and one- or two-way ANOVA (with the error term due to subject) were used to compare data for both behavior and electrophysiological measures (maximum power and maximum ITC) for high- and low-value licks, blocked versus interleaved licks, and high-intermediate-low licks. For significant ANOVAs, the error term was removed and Tukey's post hoc tests were performed on significant interaction terms for multiple comparisons. Descriptive statistics are reported as mean ± SEM, unless noted otherwise.

### Data analysis: behavior

All rats were first run for at least five standard sessions in the standard SVLT with differences in concentration (16% and 4% wt/vol). Rats have been shown to acquire incentive contrast effects in the SVLT after this duration of training (*Parent et al., 2015a*). For the blocked-interleaved and three reward tasks, rats were tested after extensive experience in the SVLT and after two 'training' sessions with the blocked-interleaved and three reward designs. The electrophysiological recordings reported here were from the animals' third session in each task.

Behavioral measures included total licks across the session, the duration and number of licking bouts, and the median inter-lick intervals (inverse of licking frequency). Bouts of licks were defined

as having at least three licks within 300 ms and with an inter-bout interval of 0.5 s or longer. Bouts were not analyzed in the blocked-interleaved task; due to the unique structure of the task, bouts were all shortened by default due to a constantly changing reward in the interleaved phase of the task. While bouts of licks were reported in most tasks, electrophysiological correlates around bouts were not analyzed because there were often too few bouts (specifically for the low-lick conditions) in each session to deduce any electrophysiological effects of reward value on bout-related activity.

For analyzing lick rate, inter-lick intervals during the different types of rewards were obtained, and then the inverse of the median inter-lick interval provided the average lick rate in Hertz. Any inter-lick interval greater than 1 s or less than 0.09 s was excluded from the analysis. For licks during the randomly interleaved portion of the blocked-interleaved task, more than two licks in a row were needed to calculate lick rate. To analyze behavioral variability of licks, we used coefficient of variation (ratio of the standard deviation to the mean) on high- and low-value inter-lick intervals that occurred within bouts.

In some experiments, imbalances were apparent in measures of total licks and lick rate. This was due in part to our only calculating inter-lick intervals that were less than 1 s and consisted of runs of at least two consecutive licks. As a result, some licks that were detected were not included in the quantitative measures of lick rate (e.g. two licks that occur 15 s apart from each other). Isolated licks occur in the behavioral design used in our studies when rats sample fluid from the spout during periods when low-value fluid is available and then do not engage in persistent licking.

Total licks and lick rate are therefore distinct measures and will not always be coupled, especially because licks occur in bursts. Rats strongly engage when the higher value fluid is available in the blocked condition and alternatively will lick more sporadically and will default to sampling the fluid and not maintain engagement when the low-value fluid is available. However, the rate of the licks, in said bouts or bursts, was higher overall during the interleaved parts of the tests sessions. Why this happened is not clear, but one interpretation is that rats are not suppressing or sampling the options anymore in the interleaved portion but are instead maintaining engagement in the task during the interleaved portion of the task when reward identity is unpredictable.

## Data analysis: LFPs

Electrophysiological data were first analyzed in NeuroExplorer (http://www.neuroexplorer.com/), to check for artifacts and spectral integrity. Subsequent processing was done using signal processing routines in GNU Octave. Analysis of LFPs used functions from the EEGLab toolbox (*Delorme and Makeig, 2004*) (ERSP and ITC) and the signal processing toolbox in GNU Octave (the peak-to-peak function was used to measure event-related amplitude). Circular statistics were calculated using the circular library for R. Graphical plots of data were made using the matplotlib and seaborn library for Python. Analyses were typically conducted in Jupyter notebooks, and interactions between Python, R, and Octave were implemented using the rpy2 and oct2py libraries for Python.

To measure the amplitude and phase of LFP in the frequency range of licking, LFPs were bandpass-filtered using eeglab's eegfilt function, with a fir1 filter (*Widmann and Schröger, 2012*), centered at the rat's licking frequency (licking frequency+ inter-quartile range; typically around 4–9 Hz), and were subsequently z-scored.

Lick-field coherence used routines (e.g. sp2a_m) from the Neurospec 2.0 library (http://www.neurospec.org/) for MATLAB and GNU Octave. LFPs were low-pass filtered (100 Hz) using eegfilt.m from EEGLab. Directed coherence also used routines (e.g. sp2a2_R2.m) from Neurospec 2.0. LFPs were low-pass filtered (100 Hz) using eegfilt.m from EEGLab. The following parameters were used for lick-field coherence and directed coherence: segment power = 10 (1024 points, frequency resolution: 0.977 Hz), Hanning filtering with 50 % tapering, and line noise removal for the LFPs at 60 Hz. Analyses focused on frequencies below 30 Hz based assessments of power spectra computed by the Neurospec library as part of this analysis.

For ITC and ERSP, LFP data was preprocessed using eeglab's eegfilt function with a fir1 filter and was bandpass-filtered from 0 to 100 Hz. For group summaries, ITC and ERSP matrices were z-scored for that given rat after bandpass-filtering the data. Peri-lick matrices were then formed by using a pre/post window of 2 s on each side, and the newtimef function from the eeglab toolbox was used to generate the time-frequency matrices for ITC and ERSP up to 30 Hz.

Since most of the lick counts from the SVLT are generally imbalanced (with a greater number of licks for high- versus low-value rewards), we used permutation testing to perform analyses on amplitude and phase-locking in these studies. Licks were typically downsampled to match the lower number of licks; 80 % of the number of lower value licks were randomly chosen from each session. For example, if a rat emitted 400 licks for the high concentration sucrose and 200 licks for the low concentration sucrose, then 160 licks would be randomly chosen from each of data type to compare the same number of licks for each lick type. This permutation of taking 80 % of the licks was re-sampled 25 times and spectral values were recalculated for each permutation. The maximum ITC value was obtained through calculating the absolute value of ITC values between 2 and 12 Hz within a ~ 150 ms window (+1 inter-lick interval) around each lick. The maximum ERSP value was also taken around the same frequency and time window. Then, the average maximum ITC or ERSP value (of the 25 × resampled values) for each LFP channel for each rat was saved in a data frame, and each electrode's maximum ITC and ERSP value for each type of lick (high-value or low-value lick) were used in the ANOVAs for group summaries. Group summary for the peak-to-peak ERP size recorded the average difference between the maximum and minimum ERP amplitude across all frequencies, using +1 inter-lick interval window around each lick. The mean ERP size for each electrode for each rat was used in the ANOVAs for group summaries. These analyses were performed for all behavioral variations.

## Acknowledgements

We thank Wambura Fobbs, Alexxai Kravitz, Catherine Stoodley, Steve Wise, and Samantha White for helpful feedback on the manuscript.

## Additional information

### Funding

| Funder | Grant reference number | Author |
| --- | --- | --- |
| National Institute on Drug Abuse | DA046375 | Mark Laubach |
| National Science Foundation | GRFP | Linda M Amarante |

The funders had no role in study design, data collection and interpretation, or the decision to submit the work for publication.

### Author contributions

Linda M Amarante, Conceptualization, Data curation, Formal analysis, Investigation, Methodology, Project administration, Resources, Software, Validation, Visualization, Writing - original draft, Writing - review and editing; Mark Laubach, Conceptualization, Data curation, Formal analysis, Funding acquisition, Investigation, Methodology, Project administration, Resources, Supervision, Validation, Visualization, Writing - original draft, Writing - review and editing

### Author ORCIDs

Linda M Amarante http://orcid.org/0000-0002-3592-7346
Mark Laubach http://orcid.org/0000-0002-2403-4497

### Ethics

All procedures carried out in this set of experiments were approved by the Animal Care and Use Committee at American University (Washington, DC). The approved protocol number is 1710. All procedures conformed to the standards of the National Institutes of Health Guide for the Care and Use of Laboratory Animals. All efforts were taken to minimize the number of animals used and to reduce pain and suffering.

### Decision letter and Author response

Decision letter https://doi.org/10.7554/eLife.63372.sa1
Author response https://doi.org/10.7554/eLife.63372.sa2

# Additional files

## Supplementary files
• Transparent reporting form

## Data availability
Code, raw data, and summaries of grouped data are available on the Open Science Framework.

The following dataset was generated:

| Author(s) | Year | Dataset title | Dataset URL | Database and Identifier |
|-----------|------|---------------|-------------|------------------------|
| Amarante L, Laubach M | 2021 | Coherent theta activity in the medial and orbital frontal cortices encodes reward value | https://osf.io/de78t/ | Open Science Framework, de78t |

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
