## [Decision Letter]

**Acceptance summary:**

In this study, Amarante and Laubach examined the role of theta oscillations in the medial frontal and orbital frontal cortices in processing reward-related information that guides consummatory behavior. Using in-vivo electrophysiology, they recorded local field potentials in awake behaving rats as they performed a shifting-values licking task, in which the delivery of fluid reward was triggered by licking a delivery spout. Reward values (sucrose concentration or fluid volume) varied in either block-wise or interleaved fashion to determine the effects of shifting value contingencies on behavior and neural activity. The main results show that theta phases time-locked to individual licks reflected differences in reward expectancies, and directional analyses further suggest an influence of medial frontal cortex on orbitofrontal cortex during consummatory behavior. Reviewers agreed that the results are convincing and represent important new insights into the role of frontal cortex theta oscillations in motivated behavior.

**Decision letter after peer review:**

Thank you for submitting your article "Rhythmic activity in the medial and orbital frontal cortices tracks reward value and the vigor of consummatory behavior" for consideration by *eLife*. Your article has been reviewed by 3 peer reviewers, one of whom is a member of our Board of Reviewing Editors, and the evaluation has been overseen by Kate Wassum as the Senior Editor. The reviewers have opted to remain anonymous.

The reviewers have discussed the reviews with one another and the Reviewing Editor has drafted this decision to help you prepare a revised submission.

We would like to draw your attention to changes in our revision policy that we have made in response to COVID-19 (https://elifesciences.org/articles/57162). Specifically, we are asking editors to accept without delay manuscripts, like yours, that they judge can stand as *eLife* papers without additional data, even if they feel that they would make the manuscript stronger. Thus the revisions requested below only address analyses, clarity, and presentation.

Summary:

Amarante and Laubach examined the role of theta oscillations in the medial frontal (MFC) and orbital frontal (OFC) cortices in processing reward-related information that guides consummatory behavior. Using in vivo electrophysiology, they recorded local field potentials in awake behaving rats performing a shifting values licking task, in which the delivery of fluid reward was triggered by licking the delivery spout. Reward values (sucrose concentration or fluid volume) varied in either block-wise or interleaved fashion to determine the effects of shifting values on behavior and neural activity. The main results show that theta phase alignments time-locked to licks reflected differences in reward expectancies. Under the simplest conditions, when rewards magnitudes are easy to anticipate, theta phase alignment in MFC and OFC show largely similar entrainment to licking behavior, increasing on blocks where more valuable outcomes were expected. However, when three levels of reward were introduced, so that the intermediate was the best in some sessions and the worst in others, lick rates reflected the relative value of the rewards whereas theta oscillations in MFC were more closely aligned with absolute reward values and those in OFC reflected reward changes in a manner that was less clearly interpretable. These results are interpreted as evidence that MFC theta coherence relates to response vigor.

Overall, the reviewers agreed that the study was well designed and carefully executed. However, there were a number of methodological points that require clarification, and there were concerns about the interpretation of some results. Specific points under each of these categories are elaborated below.

Essential revisions:

1. First, there were questions about the methods used to analyze behavior on the interleaved blocks. It appears that the rodents made nearly twice as many licks in the blocked versus interleaved context, yet the lick rates are largely equivalent, with the timing of the task conditions fixed to 3 minutes (i.e. Figure 5A vs. B). This suggests there must be some differences in timing that lead to this result. Please clarify.

Because the behavior in these blocks was not analyzed in more detail, it's unclear what processes the LFPs recorded in either region might reflect. A stronger attempt to define what happens in the interleaved blocks would be helpful here. Specifically, could the lack of effects in OFC be due to unaccounted for fluctuations in the lick rates on interleaved blocks or at the beginning of the 3-reward task? For instance, OFC might be tracking a learning process that changes more dynamically than block-by-block. It may be fruitful to split the block analyses in terms of a moving average, or early versus late portions, that reflect the expected value of licking at different points in the block. Other possibilities include assessing whether the overall behavior is consistent with matching, so there are trials where behavior looks like they anticipate a high reward and some where they anticipate a low reward, versus whether all trials look like the relatively low-reward state? Or if there are previous trial effects that look like evidence of learning, even though the outcomes are randomized?

2. Second, it appears that the frequencies included in the LFP bandpass varied, but it isn't clear whether this varied by rat or by block type. Please clarify this as well. If the latter is the case, this is a potential confound with differences in lick rates across blocks that needs to be addressed.

3. Regarding the absolute encoding of reward value, to determine whether this is encoded all combinations of contrasts would ideally be examined within the same session. The illustration of "absolute encoding" in Supp. Figure 3, like Figure 7, compares 8% to 4% and then to 16% sucrose in different blocks. However, 16% is not compared to 4% within the same session. What if, in the same session, 16% was compared to 4%, and the resulting MFC ITC values were more similar as they were to the SVLT task (16% = ~7.25 vs. 4% = ~6) or the Blocked-Interleaved task (16% = ~6.25 vs. 4% = ~5.5)? Without including them as a block within the same session, it is difficult to make assumptions about absolute encoding if not all contrasts are available to the animal.

4. In some instances, the OFC data demonstrate similar trends to the MFC data but are not statistically significant (e.g., Figure 7C vs. Figure 7F, etc.). However, there are more MFC electrodes (n = 160 from 10 rats) than OFC electrodes (n = 96 from 6 rats). Can the authors comment as to whether the OFC is under-powered? Or more variable (e.g., SEM in Figure 7C vs. Figure 7F)?

Regarding interpretation:

5. One of the more interesting results of the paper is that ITC does not follow with the overall pattern of lick rate across trials/blocks. That is, MFC signals are described as reflecting absolute value, whereas lick rates are more consistent with relative value. However, the interpretation is described as MFC coding response vigor. If response vigor is operationally defined by lick rate, how does this interpretation follow? In addition, please explain the operational definitions and differences between task engagement and response vigor, as they do not necessarily seem to be independent of one another.

6. Are the putative lick-entrained oscillations, particularly in MFC, fully dissociated from the licks themselves? The discussion notes that licking can be rhythmic with a similar frequency as the neural rhythms analyzed here. The authors suggest that the lick-entrained theta should not be interpreted as relating directly to the execution of a lick because the patterns across trials/blocks don't match those of lick rates. Therefore, they emphasize indirect comparisons between patterns of behavior across trial types and patterns of LFP activity across trial types, when a more direct measure would be more compelling. Can the authors relate the two types of measures in another way? For instance, further analyses might assess whether lick frequency is strongly correlated (e.g., Pearson's) with lick-coupled ITC across trials in any of the behavioral tasks for the MFC or the OFC. From another perspective, the total licks in a session or number of lick bouts are measures that are pooled over time, and it's possible that they don't quite capture the pattern of licks occurring in the peri-lick window in which the neural data were analyzed. To address this, licking could be aligned to the same lick times that the neural data were aligned to, and show whether there continues to be subtle divergence between behavior and neural activity.

7. Results reveal that theta oscillation phase, but not power, has a relationship to licking behavior. Can the authors describe the functional significance of phase coordination vs. power changes in these cortical networks? And speculate on why phase alignment appears to be a more important measure than differences in power? Relatedly, are these oscillations thought to be locally generated within the MFC or OFC networks?

8. Pp. 27, lines 479-481: Please explicitly describe the ways in which the present data support the hypothesis that the MFC represents action-outcome relationships, while the OFC represents stimulus-outcome relationships.

---

## [Author Response]

Essential revisions:1. First, there were questions about the methods used to analyze behavior on the interleaved blocks. It appears that the rodents made nearly twice as many licks in the blocked versus interleaved context, yet the lick rates are largely equivalent, with the timing of the task conditions fixed to 3 minutes (i.e. Figure 5A vs. B). This suggests there must be some differences in timing that lead to this result. Please clarify.

In some experiments, imbalances were apparent in measures of total licks and lick rate. This was due in part to our only calculating inter-lick intervals that were less than 1 second and consisted of runs of at least 2 consecutive licks. As a result, some licks that were detected were not included in the quantitative measures of lick rate (e.g. two licks that occur 15 seconds apart from each other). Isolated licks occur in the behavioral design used in our studies when rats sample fluid from the spout during periods when low value fluid is available and then do not engage in persistent licking.

“Total licks and lick rate are therefore distinct measures and will not always be coupled, especially because licks occur in bursts. Rats strongly engage when the higher value fluid is available in the blocked condition and alternatively will lick more sporadically and will default to sampling the fluid and not maintain engagement when the low value fluid is available. […] Why this happened is not clear, but one interpretation is that rats are not suppressing or sampling the options anymore in the interleaved portion but are instead maintaining engagement in the task during the interleaved portion of the task when reward identity is unpredictable.”

The text above was added to the Methods section on page 33 in the revised manuscript.

Because the behavior in these blocks was not analyzed in more detail, it's unclear what processes the LFPs recorded in either region might reflect. A stronger attempt to define what happens in the interleaved blocks would be helpful here. Specifically, could the lack of effects in OFC be due to unaccounted for fluctuations in the lick rates on interleaved blocks or at the beginning of the 3-reward task? For instance, OFC might be tracking a learning process that changes more dynamically than block-by-block. It may be fruitful to split the block analyses in terms of a moving average, or early versus late portions, that reflect the expected value of licking at different points in the block. Other possibilities include assessing whether the overall behavior is consistent with matching, so there are trials where behavior looks like they anticipate a high reward and some where they anticipate a low reward, versus whether all trials look like the relatively low-reward state? Or if there are previous trial effects that look like evidence of learning, even though the outcomes are randomized?

Thank you for this feedback. Data analysis is challenging in this simple behavioral procedure, which lacks trial structure and by its nature encourages rats to suppress licking when the lower value option is available. Nevertheless, we tried a number of session wide splits, moving windows, and sequential analyses to better understand results from the blocked-interleaved sessions. Most analyses were of limited utility, especially those that examined sequential effects of value on licks, due to the low number of low-value licks.

“Sessions were split into sequential 30 sec windows and the various measures of licking behavior were plotted. […] Licks for the lower value fluid increased starting from the onset of the interleaved part of the session.”

The text above has been added to the manuscript as well as Supplementary Figure 3, and noted in the text on page 11. If the reviewers and editors would like more extensive summaries for the moving window analysis, we can easily add them to the manuscript.

As reported in our 2017 paper on LFP fluctuations in the MFC in the base version of the task, there were no effects of two- or three-way session splits on the behavioral or neural measures. Rats licked in a similar manner across the portion of the session with interleaved rewards and various measures of LFC activity (power, phase locking) were stable over those periods of the sessions. Plots can be added as supplemental figures to document these negative findings, if the reviewers and editors would like to have them in the manuscript.

Sequential analyses of licks were also attempted. These included evaluating behavioral (inter-lick intervals) and neural measures (lick-field coherence) for licks that were preceded by the same or other value lick. No results were significant from these analyses, and this was likely due to a major imbalance in the number of high and low value licks. There were fewer low value licks overall and so many fewer sequences of low-high (last lick low and current lick high) and high-low licks compared to high-high licks. These results were examined using traditional statistics and also estimation statistics, and were generally underwhelming.

2. Second, it appears that the frequencies included in the LFP bandpass varied, but it isn't clear whether this varied by rat or by block type. Please clarify this as well. If the latter is the case, this is a potential confound with differences in lick rates across blocks that needs to be addressed.

All LFPs were low-pass filtered at 100 Hz for all analyses. The only analysis that used rat-specific time windows was the peak-to-peak analysis, which constricted the data analysis window to be within +/- 1 median inter-lick interval (calculated for each rat’s licking data).

Indeed, given that we used each rat’s median ILI as a range for the analysis windows in the peak-to-peak analysis, there will naturally be a slight variation in each rat’s analysis window. The intent was to use each individual rat’s behavior to drive the analysis windows instead of one standard window over all rats. To confirm that there is minimal spread of this data analysis window and that the analysis window of +/- 1 ILI was appropriate across rats, we calculated each animal’s ILI range from the MFC shifting values concentration sessions (all rats used in Figure 1D for concentration sessions in dark blue). The median inter-lick interval of all rats is 150.03 msec. There is only a 30.675 msec difference in the maximum to the minimum inter-lick interval across all rats (maximum=170.03 msec; minimum ILI = 139.35 msec).

3. Regarding the absolute encoding of reward value, to determine whether this is encoded all combinations of contrasts would ideally be examined within the same session. The illustration of "absolute encoding" in Supp. Figure 3, like Figure 7, compares 8% to 4% and then to 16% sucrose in different blocks. However, 16% is not compared to 4% within the same session. What if, in the same session, 16% was compared to 4%, and the resulting MFC ITC values were more similar as they were to the SVLT task (16% = ~7.25 vs. 4% = ~6) or the Blocked-Interleaved task (16% = ~6.25 vs. 4% = ~5.5)? Without including them as a block within the same session, it is difficult to make assumptions about absolute encoding if not all contrasts are available to the animal.

It has been difficult to make conclusions about absolute encoding with using across-session data. The three-reward task actually was initially designed to have all combinations of reward presented to the animal, as Block 1: 8/4%; Block 2: 16/8%; Block 3: 16/4%. We opted to not include the third Block in our original analyses due finding that rats emitted lower numbers of licks, specifically for the 4% sucrose. Typically, we aimed for at least ~50 licks for electrophysiological analyses, and all of the low lick counts were under 50 licks in the block (and a few rats only emitted one or two licks total, meaning we could not perform electrophysiological analysis). This could be due to satiety in the task, although all but three rats emitted many (over 50, upwards of 300+) licks for the Block 3 16% sucrose reward.

However, it may be fruitful to include this data, such that at least the Block 3 16% data can be compared with the Block 2 16% data. We went back and included Block 3 behavior measures (total licks and lick rate) and electrophysiological data for the high-value (16%) licks, which is shown in the attached figure.

In comparing lick counts, there was a significant effect of lick type on the total number of licks emitted (F(5,51)=28.49, p<0.001). Post-hoc testing revealed a significant decrease in the total number of licks emitted for the high value (16%) sucrose in Block 3 as opposed to licks for high value sucrose in Block 2 (p<0.001), although Block 3 high value lick counts were not significantly different from the Intermediate (8%) Block 2 lick counts (p=0.88). There was a significant difference in licks emitted for Block 2 versus Block 3 high value sucrose (p<0.005). Additionally, lick counts were quite low for Block 3 low value sucrose and therefore were omitted from subsequent electrophysiological analyses. Lick rate was also re-analyzed including Block 3 high value licks, but lick rate for low value (4%) licks could not be analyzed due to a low number of licks not passing criteria. Lick rate for Block 3 high value licks were not significantly different from Block 2 high value licks (p=0.99), nor from Block 2 intermediate licks (p=0.83).

In MFC, Block 3 High value ITC values [95% CI: 0.716, 0.757] were not significantly different from Block 2 High value ITCs (p=0.113) [95% CI: 0.671, 0.715]. Block 3 High value ITCs were significantly increased from both intermediate (8%) value ITCs in Block 1 [95% CI: 0.619, 0.672] and Block 2 [95% CI: 0.610, 0.658] (p<0.001 for both).

In OFC, using the same criteria of >50 licks, one rat was excluded. In the remaining animals, Block 3 High value ITCs [95% CI: 0.540, 0.611] were not significantly different from Block 2 High value ITCs (p=0.789) [95% CI: 0.497, 0.580]. Block 3 High value ITCs were significantly increased from Block 2 Intermediate value ITCs (p=0.025) [95% CI for intermediate Block 2: 0.450, 0.535], but Block 3 high value ITCs were not significantly different from Block 1 intermediate ITCs [Block 1 intermediate 95% CI: 0.505, 0.584]. The other comparisons (Block 2 intermediate versus Block 2 high value) were not significant (p=0.339).

We added new Supplementary Figure 5. We have also emphasized in the text (page 17, and also Supplementary Figure 4) that our “standard” / operational definition for absolute value is that there would be no significant difference between the same value reward across blocks, as suggested in our hypotheses in the supplement as well. We also changed the previous section labeled “Three Reward Task: Neural evidence for effects of absolute, not relative, reward value” to “Three Reward Task: Neural activity does not reflect relative reward value encoding”.

4. In some instances, the OFC data demonstrate similar trends to the MFC data but are not statistically significant (e.g., Figure 7C vs. Figure 7F, etc.). However, there are more MFC electrodes (n = 160 from 10 rats) than OFC electrodes (n = 96 from 6 rats). Can the authors comment as to whether the OFC is under-powered? Or more variable (e.g., SEM in Figure 7C vs. Figure 7F)?

Comparisons between MFC and OFC, especially for the data in the original Figure 7, were challenging for two reasons. First, there were differences in total sampling due to the project originally focusing on MFC and moving to OFC as the work progressed. The difference in total number of recordings and fewer animals with data from OFC does suggest an underpowered analysis. Second, the MFC areas are somewhat less complex anatomically, with most recordings in the rostral prelimbic and medial orbital regions, which are generally similar in terms of inputs and outputs, and mostly from the deep layers of the cortex. By contrast, the OFC recordings came from three distinct orbital regions (AId, LO, DLO) and were more scattered over layers.

Quantitatively, we re-ran the analyses for the RRV sessions and the ITC measure from MFC and OFC (original Figure 7C vs 7F). Since there was a lower *n* in the OFC groups, we downsampled the data to 90% of the *n* in the OFC group and calculated statistical measures for 1000 randomly resamples. Downsampling resulted in equivocal results for ITC differences in both MFC and OFC. For this reason, we have revised our summary of findings associated with the original Figure 7 to emphasize results from MFC and we now state that results on OFC were equivocal, possibly due to having undersampled the region in the RRV experiment. We modified the text to state: “However, these findings should be interpreted in the light of uneven sampling between areas, with fewer recordings done in the OFC. It is therefore possible that our results are underpowered for the OFC and new experiments could reveal an alternative interpretation.” (page 17).

We also changed the subtitle for the section reporting results from the Blocked-Interleaved experiment to “Blocked-Interleaved Task: MFC rhythmicity tracks response vigor.” The revised subtitle does not comment on unclear findings from the OFC.

Importantly, the uneven sampling between areas should not discount the observation that MFC tracks lick rate (vigor) in the Blocked-Interleaved experiment or the absolute value of the rewarding fluids in the Relative Reward Value experiment.

Regarding interpretation:5. One of the more interesting results of the paper is that ITC does not follow with the overall pattern of lick rate across trials/blocks. That is, MFC signals are described as reflecting absolute value, whereas lick rates are more consistent with relative value. However, the interpretation is described as MFC coding response vigor. If response vigor is operationally defined by lick rate, how does this interpretation follow? In addition, please explain the operational definitions and differences between task engagement and response vigor, as they do not necessarily seem to be independent of one another.

Our interpretations were based on operationally defining response vigor as lick rate (measured as the reciprocal of the inter-lick interval) and engagement as total licks per session. Fast licking is vigorous. A rat could lick vigorously and be either consistently or inconsistently engaged in the task. For example, the rat could emit licks in briefer bouts or with longer inter-bout intervals, leading to less time spent licking and thus reduced engagement. An engaged rat would spend more time licking over the test session than a less engaged rat, and could lick more or less vigorously.

Interpretational statements were added for lick counts as a proxy for engagement and lick rate as a proxy for vigor (last paragraph on page 33 and Figure 5). Furthermore, to support the idea that lick counts and licking rate are generally independent measures, we used Spearman rank correlation to examine the relationship between lick rate and total licks. The correlation statistic was 0.44242 (p=0.20042). This finding was added to the revised manuscript on page 12.

6. Are the putative lick-entrained oscillations, particularly in MFC, fully dissociated from the licks themselves? The discussion notes that licking can be rhythmic with a similar frequency as the neural rhythms analyzed here. The authors suggest that the lick-entrained theta should not be interpreted as relating directly to the execution of a lick because the patterns across trials/blocks don't match those of lick rates. Therefore, they emphasize indirect comparisons between patterns of behavior across trial types and patterns of LFP activity across trial types, when a more direct measure would be more compelling. Can the authors relate the two types of measures in another way? For instance, further analyses might assess whether lick frequency is strongly correlated (e.g., Pearson's) with lick-coupled ITC across trials in any of the behavioral tasks for the MFC or the OFC. From another perspective, the total licks in a session or number of lick bouts are measures that are pooled over time, and it's possible that they don't quite capture the pattern of licks occurring in the peri-lick window in which the neural data were analyzed. To address this, licking could be aligned to the same lick times that the neural data were aligned to, and show whether there continues to be subtle divergence between behavior and neural activity.

This is an excellent question and one that required extensive new analysis to address. We considered the methods that were suggested. None of them address the correlation between licks and LFC fluctuations as time series, and this seems to be the most direct way to answer the question.

Given this feedback and our previous experiences analyzing data from spike trains and LFPs, we realized that methods used for spike-field coherence could be used to address these issues, treating licks as spikes. Such lick-field coherence (LFC) estimates phase synchrony as a function of frequency. If licks fully drive LFP fluctuations at the licking frequency, then LFC should be near 1. If licks and LFP fluctuations are fully dissociated, then LFC should be near 0. For our data, we found the extent of coupling between MFC and OFC rhythmic activity was typically between 0 and 0.5, was larger for high value licks (high concentration and large volume) compared to low value licks, and was larger for sites in MFC compared to OFC (see the new Figure 3). These results suggest that the LFP oscillations near the licking frequency are partially dissociated from licking. These new results are very helpful for understanding the impact of licking on LFPs in MFC and OFC, and will be useful in discussing our results going forward. We have asked about the extent of dissociation between licks and LFPs before, and did not have a clear answer to these questions. Thanks to your comments and these new analyses, we feel that we can now confidently answer these questions.

In working through the new analyses, we also realized that more could be said about how the fluctuations in MFC and OFC related to one another, and so using other measures spectral methods (directed coherence), we have included new results that suggest that the licking-related rhythms in MFC lead those in OFC (Figure 2). Furthermore, when we plotted coherence values anatomically, we observed that the strongest coherence levels were located in the rostral MFC (Figure 2F). These new results suggest a rostral-to-caudal gradient in the influences of MFC over OFC.

7. Results reveal that theta oscillation phase, but not power, has a relationship to licking behavior. Can the authors describe the functional significance of phase coordination vs. power changes in these cortical networks? And speculate on why phase alignment appears to be a more important measure than differences in power? Relatedly, are these oscillations thought to be locally generated within the MFC or OFC networks?

This is an important point that was not covered adequately in the previous version of our manuscript. These issues were addressed in a previous paper from our group (Amarante et al., 2017). The revised manuscript now includes an extensive discussion of these issues, on pages 23-24. In brief, the data reported in the present manuscript and published studies from our group and others suggest that the act of licking synchronizes the phase of ongoing rhythms in the MFC and OFC and that this synchronization occurs during periods of sustained increases in δ band power. Δ power varies with respiration (Lockmann and Tort, 2018), and respiration is regulated to enable sustained fluid consumption. The MFC and OFC regions that we studied are separated by a frontal agranular region that is known to control jaw opening in rats (Yoshida et al., 2009), and maybe be homologous to an area called ALM (Anterior Lateral Motor cortex) in mice.

Regarding your question about the local generation of the oscillatory activity, our 2017 study also examined this issue, for the MFC. Rats were implanted with a guide cannula in one hemisphere and an electrode array in the other, and the guide cannula was used to deliver muscimol to one hemisphere. Rats were then tested in the Shifting Values Licking task, and the neural recordings showed reductions in both event-related power and phase (Figure 8 in Amarante et al., 2018). This study demonstrates that synchronization of MFC LFP to the lick cycle depends on the MFC. Combined with the new findings added to the current manuscript, using directed coherence analysis and multivariate spectral methods, we can further suggest that the MFC leads the OFC in the timing of the licking-coupled rhythms, and might even drive processing in OFC in phase with incoming sensory information generated by licking.

8. Pp. 27, lines 479-481: Please explicitly describe the ways in which the present data support the hypothesis that the MFC represents action-outcome relationships, while the OFC represents stimulus-outcome relationships.

In the discussion, we previously had stated: “The subtle differences between the two regions follow the hypothesis that MFC represents action-outcome relationships and OFC represents stimulus-outcome relationships. MFC activity may provide the “value of the action” information to OFC, while OFC may evaluate the reward and provide feedback to MFC”.”

Our present data support this general hypothesis with MFC activity largely tracking more of the action given each respective reward, since MFC ITCs strongly reflect the expected reward value. We have reduced the statement such that at this time we cannot directly provide evidence for OFC in stimulus-outcome relationships, only that OFC does not as closely reflect the action of licking (reflected in lick rate and correlations of ITC values to licks and lick rate).

We have changed the statement to read: “The subtle differences between the two regions follow the hypothesis that these areas provide different roles during consummatory behavior. We additionally provide evidence for MFC representing action-outcome relationships, as MFC ITC activity is more strongly correlated to the action of licking and may signal information about the “value of the action.” (page 18 in the revised manuscript).